# Adrb2 controls glucose homeostasis by developmental regulation of pancreatic islet vasculature

Alexis M Ceasrine, Eugene E Lin, David N Lumelsky, Radhika Iyer, Rejji Kuruvilla*

Department of Biology, Johns Hopkins University, Baltimore, United States

**Abstract** A better understanding of processes controlling the development and function of pancreatic islets is critical for diabetes prevention and treatment. Here, we reveal a previously unappreciated function for pancreatic β2-adrenergic receptors (Adrb2) in controlling glucose homeostasis by restricting islet vascular growth during development. Pancreas-specific deletion of *Adrb2* results in glucose intolerance and impaired insulin secretion in mice, and unexpectedly, specifically in females. The metabolic phenotypes were recapitulated by *Adrb2* deletion from neonatal, but not adult, β-cells. Mechanistically, *Adrb2* loss increases production of Vascular Endothelial Growth Factor-A (VEGF-A) in female neonatal β-cells and results in hyper-vascularized islets during development, which in turn, disrupts insulin production and exocytosis. Neonatal correction of islet hyper-vascularization, via VEGF-A receptor blockade, fully rescues functional deficits in glucose homeostasis in adult mutant mice. These findings uncover a regulatory pathway that functions in a sex-specific manner to control glucose metabolism by restraining excessive vascular growth during islet development.

DOI: https://doi.org/10.7554/eLife.39689.001

*For correspondence:
rkuruvilla@jhu.edu

**Competing interests:** The authors declare that no competing interests exist.

## Introduction

Glucose homeostasis is maintained by secretion of the insulin hormone from islets of Langerhans in the pancreas. Loss or dysfunction of insulin-producing β-cells in islets results in dysregulation of blood glucose levels and leads to diabetes. β-cells receive and integrate input from multiple extra-cellular cues throughout life to control insulin secretion. An important goal in diabetes prevention and treatment is a better understanding of the molecular and cellular processes governing β-cell development and function.

β-adrenergic receptors are G-Protein-Coupled Receptors (GPCRs) that are widely expressed throughout the body and respond to adrenergic nerve-derived norepinephrine or adrenal gland-derived epinephrine to regulate diverse physiological processes including glucose metabolism. In humans, β-adrenergic agonists augment circulating insulin levels, and stimulate insulin secretion from isolated islets (*Ahrén and Scherstén, 1986*; *Lacey et al., 1993*). Adult mice with global deletion of the β2-adrenergic receptor exhibit impaired glucose tolerance and glucose-stimulated insulin secretion (GSIS) (*Santulli et al., 2012*). However, the tissue-specific functions of β-adrenergic receptors in glucose metabolism remain undefined.

Here, we identify a pancreas-specific requirement for the β2-adrenergic receptor (Adrb2) in controlling glucose homeostasis by suppressing VEGF-A production in β-cells and limiting vascular growth in islets during development. Pancreas-specific loss of *Adrb2* results in glucose intolerance and impaired glucose-stimulated insulin secretion, which surprisingly, was observed only in female mice. *Adrb2* expression in islets declines from neonatal to adult stages. Consistently, Adrb2 deletion from neonatal, but not adult, β-cells elicited metabolic defects in mice, supporting a critical role for β-cell Adrb2 during development. We provide evidence that Adrb2 acts in β-cells to suppress

VEGF-A expression and thus restrict islet vascular growth, which in turn, influences insulin synthesis and secretion. Remarkably, developmental blockade of VEGF-A signaling corrects islet hyper-vascularization in neonatal mice and rescues glucose intolerance and insulin secretion defects in adult *Adrb2* mutant mice. These findings reveal Adrb2 as a negative regulator that controls islet development and glucose metabolism by influencing bi-directional communication between islet β-cells and the vasculature.

## Results

### Adrb2 is required in neonatal β-cells for glucose homeostasis and insulin secretion in female mice

Global Adrb2 knockout mice exhibit impaired glucose tolerance and glucose-stimulated insulin secretion (GSIS) at 6 months (*Santulli et al., 2012*). However, whether Adrb2, acting specifically in the pancreas, impacts β-cell function and glucose homeostasis remains unclear. To address pancreas-specific functions of Adrb2, we crossed mice carrying a floxed *Adrb2* allele (*Adrb2^{f/f}* mice) (*Hinoi et al., 2008*) with transgenic *Pdx1-Cre* mice (*Hingorani et al., 2003*) to delete *Adrb2* in cells of the pancreatic anlage starting at embryonic stages. *Pdx1-Cre;Adrb2^{f/f}* mice (henceforth referred to as *Adrb2* cKO mice) were born at expected Mendelian frequencies, had normal body weight at birth, no gross morphological abnormalities, and survived to adulthood. Significant *Adrb2* reduction was observed in *Adrb2* cKO pancreas assessed at postnatal day 6 (P6) (*Figure 1—figure supplement 1A*). Importantly, quantitative PCR (qPCR) analysis showed that levels of other α- and β-adrenergic receptors were unaltered in *Adrb2* cKO pancreas (*Figure 1—figure supplement 1A*), indicating that pancreatic Adrb2 depletion does not elicit compensatory changes in expression of other adrenergic receptor genes. Although in *Pdx1-Cre* transgenic mice, Cre recombinase activity has been reported in the hypothalamic regions (*Song et al., 2010*), there is little *Adrb2* expression in these areas (Allen Brain Atlas, http://mouse.brain-map.org/). Additionally, the (*Tg(Pdx1-Cre^{Tuv})*) transgenic mice that we employed have not been reported to carry a human growth hormone minigene, commonly found in several Cre lines, that elicits metabolic defects (*Brouwers et al., 2014*).

To assess the role of pancreatic Adrb2 in glucose homeostasis, we evaluated metabolic parameters at the whole animal level in adult *Adrb2* cKO mice and control littermates at 2 months of age. In performing these analyses, we noted that some *Adrb2* mutant mice exhibited a glucose intolerance phenotype, while in other mutants, glucose tolerance was indistinguishable from control animals. In order to understand the basis for the conflicting results from mutant animals, we assessed glucose tolerance separately in males and females. Surprisingly, we found that only female *Adrb2* cKO mice were glucose intolerant, while male *Adrb2* cKO mice exhibited normal glucose tolerance (*Figure 1A–D*). Female *Adrb2* cKO mice also showed reduced insulin secretion during the first phase of the glucose challenge (measured 5 min after the glucose challenge), as well as dampened insulin levels in the sustained second phase (30 min after the glucose challenge) compared to same-sex control mice (*Figure 1E*). In contrast, glucose-induced insulin secretion was unaffected in male *Adrb2* cKO mice (*Figure 1F*). Consistent with previous studies in mice (*Gannon et al., 2018*; *Goren et al., 2004*; *Lavine et al., 1971*), control males showed lower glucose tolerance relative to control females (compare *Figure 1A,B versus 1C,D*), and also lower glucose-stimulated insulin secretion (compare *Figure 1E and F*). Using qPCR analyses, we found similar depletion (~90% decrease) of *Adrb2* mRNA from male and female mutant islets relative to same-sex controls (*Figure 1—figure supplement 1B*), indicating that absence of metabolic phenotypes in male mutant mice is not due to inefficient Adrb2 deletion. Both male and female *Adrb2* cKO mice showed normal insulin sensitivity (*Figure 1—figure supplement 1C and D*), suggesting that the glucose intolerance in female *Adrb2* cKO mice does not stem from defects in insulin responsiveness.

Given the unexpected sex-specific metabolic phenotypes in *Adrb2* cKO mice, we chose to mainly focus on female mice for the rest of our studies to understand the role of pancreatic Adrb2 in regulating glucose homeostasis. The reasons why male *Adrb2* cKO mice are protected from deficits in glucose homeostasis remain unclear. However, we note that even in control animals, there are sex-specific differences in islet Adrb2 expression, with the levels in male islets being significantly lower relative to females (*Figure 1—figure supplement 1B*; also see *Figure 1H*).

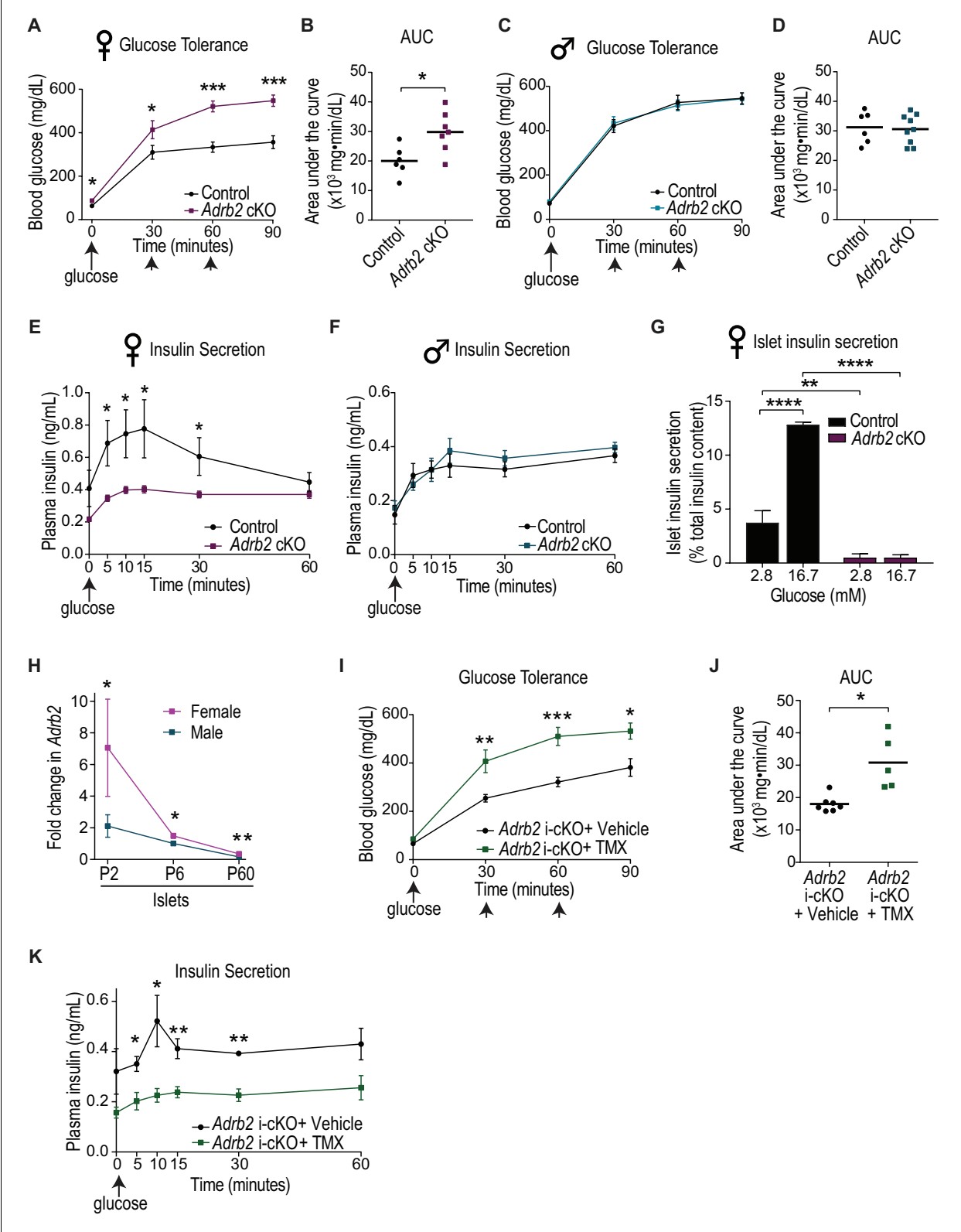

**Figure 1.** Loss of Adrb2 in neonatal β-cells results in glucose intolerance and impaired insulin secretion in female mice. (**A**) Adult (2-month-old) female *Adrb2* cKO mice have elevated fasting blood glucose and are glucose intolerant. Means ± SEM for n = 6 control and seven mutant female mice. *p<0.05, ***p<0.001, *t*-test. (**B**) Area under the curve (AUC) for glucose tolerance. *p<0.05, *t*-test. (**C**) Glucose tolerance is unaffected in male *Adrb2* cKO mice at 2 months. Means ± SEM for n = 6 control and nine mutant mice for glucose tolerance. (**D**) Area under the curve (AUC) for glucose

*Figure 1 continued on next page*

*Figure 1 continued*

tolerance in males. (**E, F**) Glucose-stimulated insulin secretion (GSIS) in vivo is reduced in female but not male *Adrb2* cKO mice. Means ± SEM for n = 6 control and eight mutant female mice; n = 5 control and six mutant male mice *p<0.05, *t*-test. (**G**) Decreased basal insulin secretion and GSIS in isolated adult female *Adrb2* cKO islets. Means ± SEM from n = 4 control and six mutant mice. **p<0.01, ****p<0.0001, two-way ANOVA with Bonferroni's post-test. (**H**) Islet *Adrb2* expression declines postnatally and is significantly lower in adult males and females compared to neonatal stages. For female islets, p<0.01, *t*-test for P60 compared to P6. For male islets, p<0.0001 for P60 compared to P6 (one sample *t*-test since male P6 values were normalized to (1). *Adrb2* levels are higher in female islets compared to males at all timepoints assessed. *p<0.05, **p<0.01, *t*-test. *Adrb2* expression in P2, P6, and P60 islets was assessed by qRT-PCR analyses and data were normalized to 18S rRNA. Results are means ±SEM and expressed as fold-change relative to P6 male islets for n = 3–5 mice/sex/age. (**I**) Neonatal β-cell-specific *Adrb2* deletion elicits glucose intolerance in mice. Neonatal *Adrb2* i-cKO mice were injected with TMX or vehicle on the day of birth and 1 day later (P0–P1), and glucose tolerance was tested when mice were 2 months old. Means ± SEM for n = 7 vehicle and 5 TMX-injected *Adrb2* i-cKO mice. *p<00.5, **p<0.01, ***p<0.001, *t*-test. (**J**) AUC for glucose tolerance. *p<0.05, *t*-test. (**K**) Neonatal β-cell-specific *Adrb2* deletion results in impaired GSIS. Means ± SEM for n = 4 vehicle and 4 TMX-injected *Adrb2* i-cKO mice. *p<0.05, **p<0.01, *t*-test.

DOI: https://doi.org/10.7554/eLife.39689.002

The following source data and figure supplements are available for figure 1:

**Source data 1.** This spreadsheet includes raw data for glucose tolerance and insulin secretion.

DOI: https://doi.org/10.7554/eLife.39689.005

**Figure supplement 1.** Adrb2 expression, insulin sensitivity and islet morphology in adult *Adrb2* cKO mice, and effects of adult β-cell-specific *Adrb2* deletion on glucose tolerance and insulin secretion.

DOI: https://doi.org/10.7554/eLife.39689.003

**Figure supplement 1—source data 1.** This spreadsheet includes raw data for Adrb2 expression, metabolic analyses, and islet morphology.

DOI: https://doi.org/10.7554/eLife.39689.004

We next asked if the altered insulin secretion in female *Adrb2* cKO mice was an islet-intrinsic defect. Thus, we measured basal insulin secretion (in response to 2.8 mM glucose), and insulin release induced by high glucose (16.7 mM) in isolated islets from control and *Adrb2* cKO mice. Both basal and glucose-stimulated insulin secretion were significantly blunted in adult female *Adrb2* cKO islets (*Figure 1G*). Together, these results reveal a pancreas-specific requirement for Adrb2 in regulating glucose homeostasis and insulin secretion specifically in female animals.

To address if glucose intolerance and impaired insulin secretion in female *Adrb2* cKO mice stemmed from defective islet formation and/or maintenance, we performed immunostaining for the islet hormone markers, insulin and glucagon, in adult animals at 2 months of age. These analyses revealed the stereotypical arrangement of insulin-producing β-cells at the core surrounded by glucagon-producing α-cells at the periphery or mantle in both control and *Adrb2* mutant islets (*Figure 1—figure supplement 1E*), suggesting that islet shape and cyto-architecture were unaffected by pancreatic *Adrb2* loss. Intriguingly, we observed an increase in insulin immunoreactivity in adult *Adrb2* cKO islets (*Figure 1—figure supplement 1E*). Enhanced insulin expression in mutant islets was confirmed by ELISA (*Figure 1—figure supplement 1F*). These results indicate that islet insulin content is increased with pancreatic Adrb2 deletion, despite a profound reduction in insulin secretion (see *Figure 1G*). Morphometric analyses revealed normal endocrine cell numbers in adult *Adrb2* mutant islets (*Figure 1—figure supplement 1G*), although the mutants had a greater distribution of smaller islets relative to control *Adrb2*^f/f mice (*Figure 1—figure supplement 1H*). Despite smaller islets, there were no significant differences in the number of EdU/insulin double-positive cells between *Adrb2* cKO and control islets (*Figure 1—figure supplement 1I,J*). The percentage of replicating β-cells in both *Adrb2* cKO and control adult islets was low (<1%), consistent with previous studies (*Puri et al., 2018*). Together, these results suggest that the metabolic phenotypes with pancreatic Adrb2 deletion do not arise from alterations in islet architecture, β-cell mass, or diminished insulin production.

In *Adrb2* cKO mice, Adrb2 would be deleted in all cells of the pancreas anlage. We next asked if Adrb2 is specifically required in β-cells to regulate insulin secretion and glucose homeostasis. Previously, Adrb2 has been reported to be enriched in β-cells at embryonic and neonatal stages, and its expression declines with age (*Berger et al., 2015*). We analyzed *Adrb2* transcript levels in neonatal islets (postnatal days 2 and 6) and in adult islets (postnatal day 60) using qPCR analyses. Consistent with previous findings (*Berger et al., 2015*), *Adrb2* expression was significantly depleted in adult islets compared to neonates (*Figure 1H*). The postnatal decline in *Adrb2* mRNA was observed in

both female and male islets. We also observed that *Adrb2* mRNA is 4-fold higher in purified β-cells relative to the non-β-cell population at postnatal day 6 (P6) (*Figure 1—figure supplement 1K*). These findings raised the possibility that Adrb2 acts in β-cells during a critical neonatal window to influence islet function.

To test the functional requirement for Adrb2 in neonatal β-cells, we crossed floxed *Adrb2* (*Adrb2^{f/f}*) mice with transgenic mice expressing tamoxifen-inducible Cre-ER fusion protein driven by the *Pdx1* promoter (*Tg(Pdx1-cre/Esr1\*)^{Dam}*) (*Gu et al., 2002*). Pdx1 expression becomes restricted to β-cells and somatostatin-expressing δ-cells after birth (*DiGruccio et al., 2016*; *Guz et al., 1995*). Therefore, we administered tamoxifen or vehicle corn oil to neonatal *Pdx1-cre/Esr1\*^{Dam};Adrb2^{f/f}* mice (henceforth referred to as *Adrb2* i-cKO mice) at the day of birth (P0) and postnatal day one (P1). Pups were allowed to grow to adulthood before testing for glucose tolerance and insulin secretion at 2 months of age. Tamoxifen injection in neonatal mice resulted in significant *Adrb2* mRNA depletion in adult islets (86% decrease, p=0.01, one sample *t*-test). Neonatal loss of Adrb2 from β-cells elicited pronounced defects in glucose tolerance and glucose-stimulated insulin secretion (*Figure 1I–K*), similar to the phenotypes observed in *Adrb2* cKO mice. Consistent with the low Adrb2 expression in adult mice, there were no significant differences in glucose tolerance or glucose-stimulated insulin secretion (GSIS) between the tamoxifen- and vehicle-injected adult *Adrb2* i-cKO mice at 6 weeks of age (*Figure 1—figure supplement 1L–O*). Together, these findings indicate that Adrb2 is primarily required in neonatal β-cells for insulin secretion and glucose homeostasis.

## Adrb2 suppresses insulin expression and islet vasculature during development

How does Adrb2 influence β-cells during islet development? Given our results that Adrb2 is largely enriched in neonatal β-cells, the similar metabolic phenotypes in *Adrb2* cKO and *Adrb2* i-cKO mice with neonatal deletion, and since tamoxifen injections in neonatal mice resulted in some pup mortality, we used the non-inducible *Adrb2* cKO mice for these and later analyses. Similar to adult mutant mice, islet formation and endocrine cell numbers were unaffected in neonatal *Adrb2* cKO mice at P6, (*Figure 2A* and *Figure 2—figure supplement 1A*), while mutants had a greater distribution of smaller islets relative to control *Adrb2^{f/f}* mice (*Figure 2—figure supplement 1B*). β-cell proliferation in neonatal islets was also unchanged by *Adrb2* deletion (*Figure 2—figure supplement 1C,D*), although we noted higher β-cell proliferation in P6 islets compared to adults as expected (see *Figure 1—figure supplement 1J*). Notably, insulin immunoreactivity (*Figure 2A*) and islet insulin content (*Figure 2—figure supplement 1E*) were increased in neonatal (P6) *Adrb2* cKO islets, similar to the findings in adult mutant islets. qPCR analyses revealed an increase in insulin transcript (*Ins2*) levels in neonatal *Adrb2* cKO islets beginning two days after birth (*Figure 2B*). Further, ultra-structural analyses revealed a pronounced increase in insulin granule density in β-cells from *Adrb2* cKO islets at P6 (*Figure 2C,D*). These results suggest that Adrb2 negatively regulates insulin expression during islet development. To ask if Adrb2 directly suppresses insulin, we stimulated cultured MIN6 cells, an insulinoma-derived cell line analogous to β-cells, with epinephrine or norepinephrine, the endogenous ligands for Adrb2, or salbutamol, an Adrb2-specific agonist. Assessment of *Ins2* transcript levels showed Adrb2 activation did not elicit any changes (*Figure 2—figure supplement 1F*). These results hint that the repressive effect of Adrb2 on insulin expression in β-cells is indirect.

Pancreatic islets are highly vascularized where endocrine cells are embedded in a dense meshwork of capillaries that allows for accurate glucose sensing, provides continuous oxygen supply to facilitate high aerobic metabolism, and ensures rapid insulin secretion into the blood stream (*Bonner-Weir and Orci, 1982*; *Eberhard et al., 2010*; *Hogan and Hull, 2017*). The intra-islet vascular network is established during embryonic development, but undergoes pronounced expansion after birth, concomitant with endocrine cell clustering and maturation (*Brissova et al., 2006*; *Johansson et al., 2006a*). We examined islet vasculature in neonatal *Adrb2* cKO mice by immunostaining for the endothelial cell-specific protein, PECAM1 (CD31). Strikingly, these analyses revealed a prominent increase in intra-islet vasculature in *Adrb2* cKO islets (*Figure 2E,F*), similar to observations of increased insulin. Intra-islet capillaries are highly fenestrated to allow for the rapid exchange of nutrients and hormones between islet-cells and the bloodstream (*Bearer and Orci, 1985*; *Henderson and Moss, 1985*). Using transmission electron microscopy, we observed altered endothelial cell morphologies with thicker cell bodies, fewer fenestrae, and more caveoloae, which are

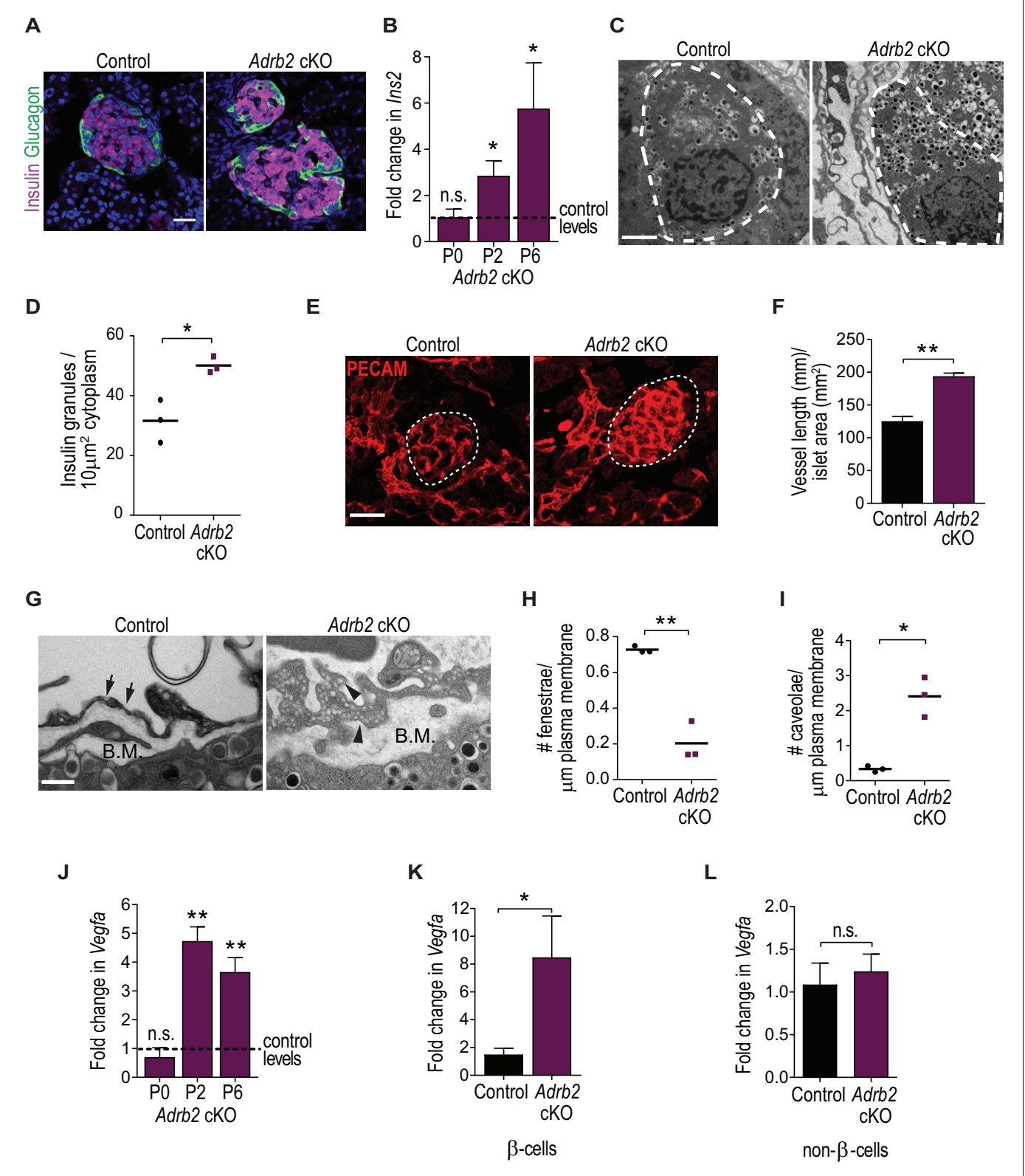

**Figure 2.** Adrb2 suppresses insulin expression and islet vasculature in neonatal mice. (**A**) Adrb2 loss increases insulin immunoreactivity in neonatal (**P6**) islets, although islet cyto-architecture is unaffected. Scale bar, 25 μm. (**B**) Enhanced *Ins2* transcript levels in neonatal *Adrb2* cKO pancreata. *Ins2* levels were assessed by qRT-PCR and normalized to 18S RNA. Results are means ±SEM for n = 4 mice per genotype for the P0 and P2 time points, and 10 mice per genotype for P6. *p<0.05, n.s. not significant, one-sample *t*-test. (**C**) Transmission electron microscopy reveals an increase in insulin granules in

*Figure 2 continued on next page*

*Figure 2 continued*

β-cells from P6 *Adrb2* cKO mice. β-cells outlined in dashed lines. Scale bar, 2 μm. (D) Quantification of cytoplasmic insulin granule density. Means for n = 3 mice per genotype. *p<0.05, *t*-test. (E) *Adrb2* cKO mice have excess intra-islet vasculature, using PECAM1 immunostaining. Islets are outlined in dashed lines. Scale bar, 50 μm. (F) Quantification of total vessel length (mm) per islet area (mm$^2$). Means ± SEM for n = 3 mice per genotype. **p<0.01, *t*-test. (G) Transmission electron microscopy shows disruptions in endothelial morphologies and vascular basement membrane in *Adrb2* cKO islets. Fenestrae (arrows) are reduced, while caveolae (arrowheads) are increased in *Adrb2* cKO islets. Basement Membrane (B.M) is expanded in mutants. Scale bar, 500 nm. (H) Quantification of fenestrae and (I) caveolae density. Means for n = 3 mice per genotype, *p<0.05, **p<0.01, *t*-test. (J) *Vegfa* transcript levels are significantly increased in the *Adrb2* cKO pancreas starting at P2. *Vegfa* levels were assessed by qRT-PCR analysis and normalized to 18S RNA. Results are means ±SEM expressed as fold-change relative to age-matched control *Adrb2^{f/f}* values. n = 3 P0 per genotype, 3 control and four mutant P2, and 5 P6 mice per genotype **p<0.01, n.s. not significant, one-sample *t*-test. (K) Increased *Vegfa* mRNA in purified β-cells from P6 *Adrb2* cKO mice. *Vegfa* levels were assessed by qRT-PCR analysis and normalized to 18S RNA. Means ± SEM and expressed as fold-change relative to control values. n = 5 control and seven mutant mice. *p<0.05, one sample *t*-test. (L) *Vegfa* transcript levels (normalized to 18S rRNA) are unchanged in non-β-cells from *Adrb2* cKO mice. Means ± SEM and expressed as fold-change relative to control values. n = 4 control and six mutant mice. n.s. not significant, one sample *t*-test.

DOI: https://doi.org/10.7554/eLife.39689.006

The following source data and figure supplements are available for figure 2:

**Source data 1.** This spreadsheet includes raw data for islet morphology and transcript changes in neonatal mice.
DOI: https://doi.org/10.7554/eLife.39689.009
**Figure supplement 1.** Islet morphology and vasculature in neonatal *Adrb2* cKO mice.
DOI: https://doi.org/10.7554/eLife.39689.007
**Figure supplement 1—source data 1.** This spreadsheet includes raw data for islet morphology and transcript changes in neonatal mice.
DOI: https://doi.org/10.7554/eLife.39689.008

plasma membrane invaginations involved in macromolecule transport across endothelial cells (*Parton and Simons, 2007*), in *Adrb2* cKO islets (*Figure 2G–I*). Islet endothelial cells also produce the vascular basement membrane, a specialized extracellular matrix consisting of collagen and laminin proteins that offer structural support, influence β-cell proliferation, and insulin gene transcription (*Kaido et al., 2004*; *Nikolova et al., 2006*; *Nikolova et al., 2007*). Ultra-structurally, we observed an increase in the thickness of the basement membrane in *Adrb2* cKO islets (*Figure 2G*), as well as enhanced immunoreactivity for Collagen IV and Laminin-411/511 (*Figure 2—figure supplement 1G*), two of the most abundant basement membrane molecules (*Kaido et al., 2004*; *Nikolova et al., 2006*). Since intra-islet capillaries have been proposed to form a scaffold to promote postnatal growth of autonomic fibers (*Reinert et al., 2014*), we performed Tyrosine Hydroxylase (TH) immunostaining to visualize sympathetic axons innervating *Adrb2* cKO and control islets. Despite the pronounced increase in intra-islet endothelial cells, sympathetic innervation in and around *Adrb2* cKO islets was indistinguishable from that in control tissues (*Figure 2—figure supplement 1H,I*). These results suggest that Adrb2 suppresses vascular growth during islet development. Since in *Adrb2* cKO mice, *Pdx1*-Cre activity is restricted to cells in the pancreatic anlage (*Hingorani et al., 2003*), we reason that islet hyper-vascularization does not arise from Adrb2 deletion in vascular cells.

To understand the basis for hyper-vascularized islets in female *Adrb2* cKO mice, we assessed expression of VEGF-A, the primary vascular mitogen in islets that is predominantly produced by β-cells (*Brissova et al., 2006*). We observed a pronounced increase (3.5 – 4.7-fold) in *Vegfa* mRNA in *Adrb2* cKO pancreas starting at two days after birth, similar to the increase in insulin expression (*Figure 2J*). To determine if excess *Vegfa* expression is β-cell specific, we assessed *Vegfa* transcript levels in purified β-cells isolated by FACS from P6 *Adrb2* cKO mice that were mated to MIP-GFP mice (where GFP is expressed under the mouse insulin promoter) (*Hara et al., 2003*). Mutant β-cells showed an 8-fold increase in *Vegfa* transcript levels compared to control β-cells (*Figure 2K*). There were no differences in *Vegfa* mRNA between control and mutant non-β-cells (*Figure 2L*). We next asked if Adrb2 activity directly regulates VEGF-A expression by treating MIN6 cells with the Adrb2 agonists, salbutamol, epinephrine, or norepinephrine. Salbutamol and epinephrine significantly suppressed *Vegfa* mRNA levels, while norepinephrine had no effect (*Figure 2—figure supplement 1J*). Together, these findings provide evidence for a developmental role for pancreatic Adrb2 activity in restricting intra-islet vascular growth by limiting VEGF-A production in β-cells.

Given sex-specific differences in *Adrb2* expression between male and female islets (*Figure 1H*), one prediction would be that lower Adrb2 expression in males would correlate with enhanced *Vegfa*

levels and intra-islet vasculature relative to females. Indeed, *Vegfa* mRNA was 2.6-fold higher in neonatal (P6) male pancreata compared to females (*Figure 2—figure supplement 1K*). PECAM1 immunoreactivity was also increased in neonatal male islets compared to females (*Figure 2—figure supplement 1L,M*). These results suggest that *Vegfa* expression and intra-islet vasculature are higher in neonatal male islets, which inversely correlate with the lower *Adrb2* expression compared to females. Adrb2 loss had no effect on *Vegfa* expression or islet vasculature in neonatal male islets (*Figure 2—figure supplement 1K–M*), suggesting that male islets are less susceptible to Adrb2 loss compared to females, similar to observations of intact glucose tolerance and insulin secretion in adult male *Adrb2* cKO mice (see *Figure 1C,D and F*).

## Loss of Adrb2 perturbs islet calcium responses and exocytosis

Our findings so far indicate that adult female *Adrb2* cKO islets have a pronounced impairment in insulin secretion, despite elevated insulin content that is manifested as early as neonatal stages. We next sought to understand the basis for the β-cell secretory dysfunction in *Adrb2* cKO mice. In β-cells, regulatory control of insulin secretion occurs at the level of glucose uptake and metabolism, β-cell plasma membrane depolarization, and insulin granule mobilization and exocytosis (*MacDonald et al., 2005*). To address mechanisms by which Adrb2 influences islet insulin secretion, we exposed islets to high potassium chloride (KCl) which depolarizes the β-cell plasma membrane and results in the opening of voltage-dependent $Ca^{2+}$ channels to promote the exocytosis of secretion-ready insulin granules (*Hatlapatka et al., 2009*), thus by-passing the need for glucose uptake and metabolism to trigger insulin release. We observed a striking decrease in KCl-induced insulin secretion in isolated *Adrb2* cKO islets, similar to the defects in GSIS in mutant islets (*Figure 3A*). These findings suggest that Adrb2 regulates step(s) in insulin secretion that are at least distal to β-cell plasma membrane depolarization.

To further probe the role for Adrb2 in GSIS, we used transmission electron microscopy to examine insulin granule docking at the plasma membrane in response to glucose. In control *Adrb2^{f/f}* mice, we observed a marked increase in insulin granule localization at the β-cell plasma membrane in response to an in vivo glucose challenge compared to the fasted state (*Figure 3B,C*). However, glucose-induced recruitment of insulin granules to the cell surface was completely suppressed in *Adrb2* cKO β-cells (*Figure 3B,C*). These results suggest that Adrb2 is necessary for glucose-dependent insulin granule positioning at the β-cell surface.

Elevation in intracellular $Ca^{2+}$ concentration is a critical determinant of insulin granule exocytosis and GSIS in β-cells (*Rorsman and Ashcroft, 2018*). Glucose-induced enhancement of β-cell electrical activity triggers an increase in cytoplasmic $Ca^{2+}$ levels, largely via influx of extracellular $Ca^{2+}$ through voltage-gated $Ca^{2+}$ channels (*Rorsman and Ashcroft, 2018*). Given the deficits in insulin granule localization at the β-cell plasma membrane and the pronounced impairment in insulin secretion upon Adrb2 loss, we asked if Adrb2 is required for glucose-dependent calcium responses in β-cells. Isolated islets were loaded with the calcium indicator dye, Fluo-4 AM, and islets were imaged while being stimulated with low glucose (2.8 mM) followed by high glucose (20 mM) and subsequent depolarization with 30 mM KCl. Control islets showed a robust increase in intracellular calcium in response to high glucose and to subsequent KCl-stimulated membrane depolarization, but in contrast, *Adrb2* cKO islets showed significantly diminished calcium responsiveness under both conditions (*Figure 3D,E*, and *Videos 1* and *2*).

To gain further insight into the mechanisms underlying dysregulated calcium responsiveness and impaired GSIS in *Adrb2* cKO islets, we examined the expression of several genes known to be involved in glycolysis, β-cell depolarization, $Ca^{2+}$ influx, and insulin granule exocytosis, by qPCR analyses of adult *Adrb2* cKO and control islets at 1.5 – 2 months of age. Notably, we observed a significant down-regulation (72% decrease) in *Cacna1c* which encodes for Cav1.2, the principal L-type voltage-gated calcium channel in β-cells (*Figure 3F*). Similar to findings for *Adrb2* loss, β-cell-specific ablation of *Cacna1c* suppresses GSIS and elicits systemic glucose intolerance in mice (*Schulla et al., 2003*). We also observed down-regulated gene expression for components of the exocytosis apparatus in β-cells, *Snap-25*, *Rph3al* (Noc2), and *Pclo* (Piccolo) (*Figure 3F*). Noc2, an effector of the Rab3 and Rab27 GTPases, is an essential component of insulin granule exocytosis (*Matsumoto et al., 2004*; *Matsunaga et al., 2017*). Piccolo is a scaffold protein that functions as a $Ca^{2+}$ sensor and links $K_{ATP}$ channels, L-type calcium channels and insulin granules into exocytosis-competent complexes (*Shibasaki et al., 2004*), whereas Snap-25 is a t-SNARE protein that is critical

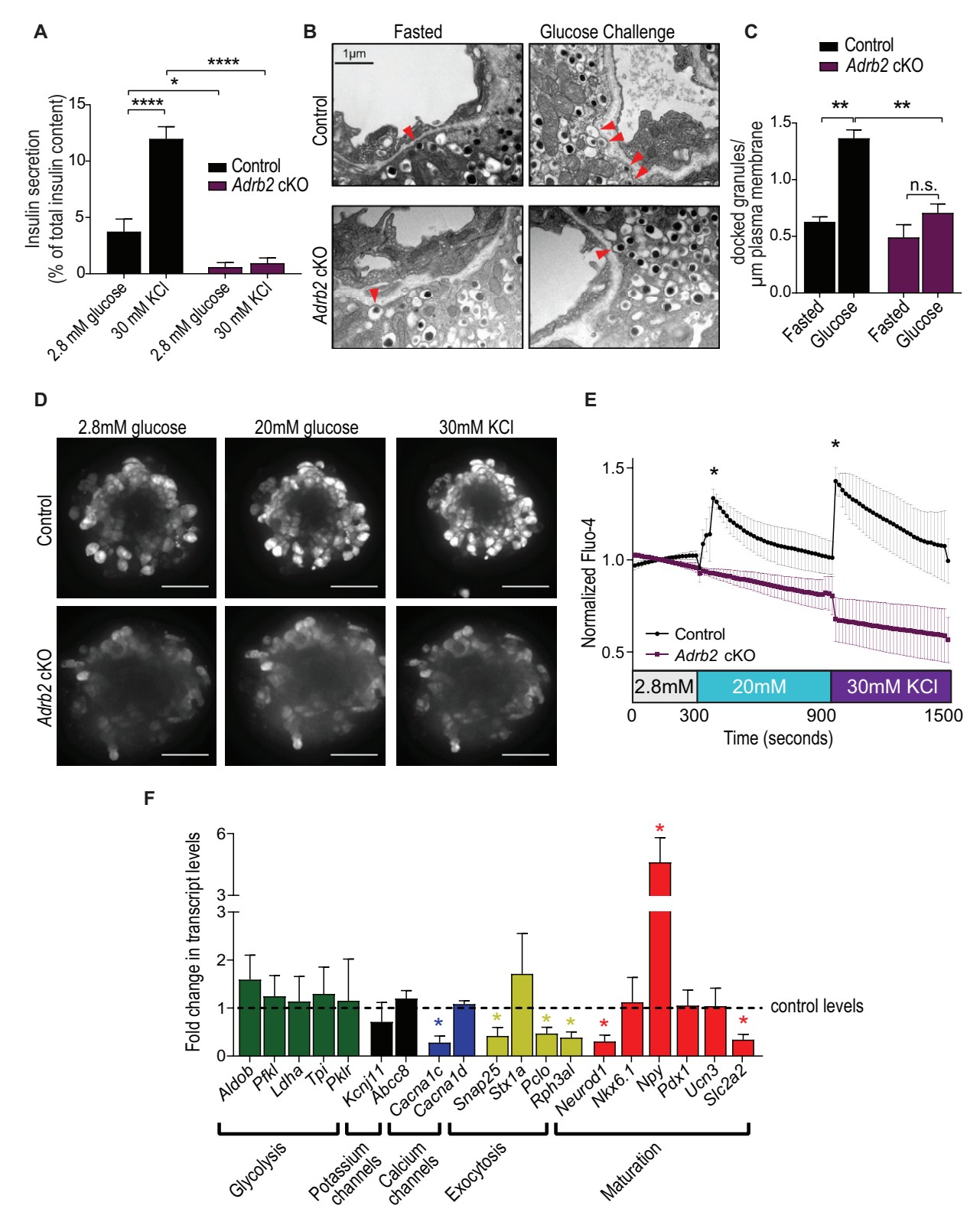

**Figure 3.** Loss of Adrb2 perturbs islet calcium responses and exocytosis. (**A**) Adrb2 is required for KCl-induced insulin secretion. Means ± SEM for n = 4 control and five mutant mice. *p<0.05, ****p<0.001, two-way ANOVA with Bonferroni's post-test. (**B**) Adrb2 is necessary for glucose-induced surface localization of insulin granules. Docked insulin granules in β-cells (within 50 nm of plasma membrane) are indicated by arrowheads. Scale bar, 1 μm. (**C**) Quantification of docked insulin granules per micron of plasma membrane. Means ± SEM from n = 3 mice each per genotype/per condition. **p<0.01,
*Figure 3 continued on next page*

*Figure 3 continued*

n.s. not significant; two-way ANOVA with Bonferonni's post-test. (**D**) Loss of Adrb2 loss impairs islet calcium responses induced by high glucose or KCl. Scale bar, 50 μm. (**E**) Quantification of normalized Fluo-4 intensities over time. Means ± SEM for n = 3 female mice per genotype, 10 cells analyzed per animal. (**F**) Decreased expression of transcripts involved in calcium signaling and insulin exocytosis in isolated islets from adult female *Adrb2* cKO mice. *Adrb2* loss also results in aberrant expression of genes involved in β-cell maturation including *Neurod1*, *Npy*, and *Slc2a2*. Transcript levels were assessed by qRT-PCR and data normalized to 18S RNA. Means ± SEM and expressed as fold-change relative to control female *Adrb2^{f/f}* values. n = 6 control and 3 – 5 mutants. *p<0.05, one sample *t*-test compared to normalized female control values.
DOI: https://doi.org/10.7554/eLife.39689.010

The following source data and figure supplement are available for figure 3:

**Source data 1.** This spreadsheet includes raw data for insulin secretion, calcium responses, and islet gene expression.
DOI: https://doi.org/10.7554/eLife.39689.012

**Figure supplement 1.** Altered expression of Glut2 and NPY in *Adrb2* cKO islets.
DOI: https://doi.org/10.7554/eLife.39689.011

for insulin granule fusion (*Wheeler et al., 1996*). Since in β-cells, *Noc2*, *Pclo* and *Snap-25* are known to be regulated by the basic helix-loop-helix (bHLH) transcription factor Neurod1 (*Gu et al., 2010*), we also assessed *Neurod1* levels in *Adrb2* cKO islets and observed a significant decrease (*Figure 3F*). The decreased *Neurod1* expression in *Adrb2* cKO islets suggests that β-cell maturation may be affected by Adrb2 deletion (*Gu et al., 2010*). To address if Adrb2 loss affects β-cell maturation, we analyzed mRNAs for additional genes associated with β-cell functional maturity including *Nkx6.1*, *Npy*, *Pdx1*, *Ucn3*, and *Slc2a2*. Of these genes, we found that *Npy* mRNA was significantly elevated in *Adrb2* cKO islets while *Slc2a2* was decreased (*Figure 3F*). *Npy* encodes for Neuropeptide Y (NPY), a hormone that is normally downregulated in islets during postnatal development (*Myrsén-Axcrona et al., 1997*), while its persistent expression in adult islets is associated with decreased β-cell responsiveness (*Gu et al., 2010*; *Rodnoi et al., 2017*). *Slc2a2* encodes for the β-cell-specific glucose transporter, Glut2, which is essential for GSIS in mature β-cells (*Thorens, 2003*). Immunostaining also revealed a marked upregulation of NPY, and diminished Glut2 expression, in adult *Adrb2* cKO islets (*Figure 3—figure supplement 1*). There were no significant differences in genes involved in glycolysis (*Aldob*, *Pfkl*, *Ldha*, *Tpi*, or *Pklr*) or encoding for potassium channel subunits found in β-cells (*Kcnj11* or *Abcc8*) between *Adrb2* cKO and control islets (*Figure 3F*). Together, these results suggest that impaired insulin secretion in *Adrb2* cKO islets arise, in part, from decreased expression of key components of the calcium regulation and exocytotic machinery in β-cells, as well as aberrant expression of specific genes associated with β-cell maturity.

## Developmental VEGF blockade rescues defects in islet morphology, insulin secretion, and glucose tolerance in *Adrb2* cKO mice

Given that islet vasculature has been postulated to impact adult islet function (*Brissova et al., 2014*; *Brissova et al., 2006*; *Cai et al., 2012*), we next asked if early developmental phenotypes in VEGF-A expression and islet hyper-vascularization in *Adrb2* cKO mice are causal for the later glucose intolerance and insulin secretion defects. If so, then remedying the neonatal hyper-vascularization should rescue the metabolic phenotypes in adult *Adrb2* cKO mice. To test this prediction, we sought to attenuate

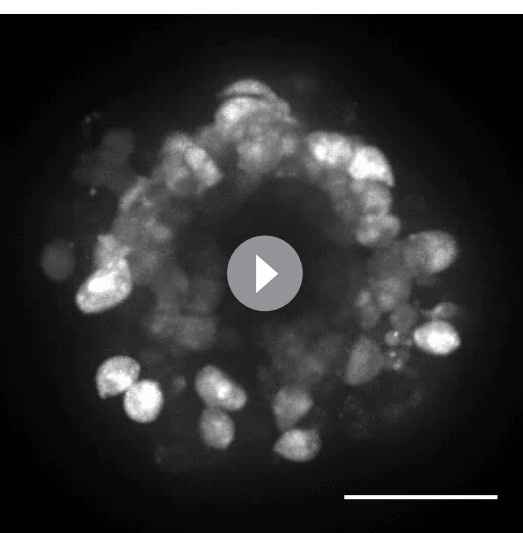

**Video 1.** Glucose- and KCl-induced calcium responses in a control islet. Representative video (15 frames per second) of a Fluo-4 AM-loaded islet isolated from a female control (*Adrb2^{f/f}*) mouse. Both high glucose and KCl elicit a robust increase in islet intracellular calcium levels.
DOI: https://doi.org/10.7554/eLife.39689.013

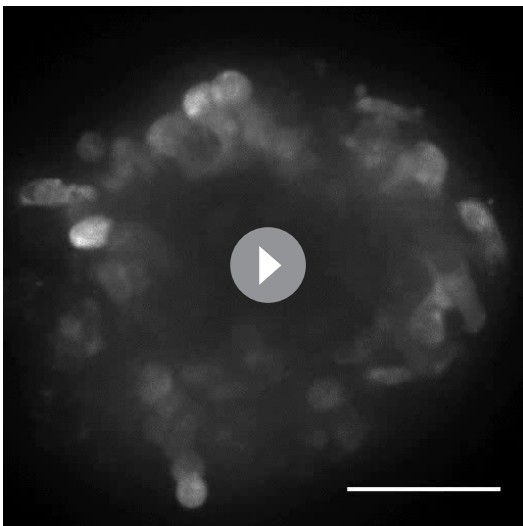

**Video 2.** Diminished calcium responses to high glucose and KCl in an *Adrb2* cKO islet. Representative video (15 frames per second) of a Fluo-4AM loaded islet isolated from a female *Adrb2* cKO mouse. Islet calcium levels are not elevated by high glucose or 30 mM KCl in the absence of Adrb2.
DOI: https://doi.org/10.7554/eLife.39689.014

VEGF-A signaling during development by injecting a neutralizing monoclonal antibody against VEGFR-2 (KDR or Flk-1), the primary VEGF-A receptor expressed in intra-islet endothelial cells (*Brissova et al., 2006*; *Kim et al., 2011*). Since developmental anomalies in *Adrb2* cKO mice manifested at postnatal day 2 (P2), mice were injected with VEGFR2 blocking antibody every three days, starting at birth (postnatal day 0), and mice were either harvested for morphological analyses at P6, or following a final injection at P6, allowed to grow into adulthood and metabolic parameters were assessed at 2 months of age. Antibody-injected *Adrb2* cKO mice survived to adulthood, had normal body weight, and did not exhibit any gross morphological abnormalities. VEGFR2 blocking antibody administration corrected the excessive islet vascularization in neonatal *Adrb2* cKO mice (*Figure 4A,B*), although it had no effect on elevated *Vegfa* expression (*Figure 4—figure supplement 1A*), as expected. VEGFR2 neutralization also normalized *Ins2* transcript levels in neonatal pancreas (*Figure 4C*), suggesting that enhanced insulin expression is a consequence of aberrant VEGF-A signaling and perturbed islet vasculature in *Adrb2* cKO mice. Remarkably, administration of the VEGFR2 blocking antibody during the first week of birth fully rescued the deficits in glucose tolerance and islet GSIS in adult *Adrb2* cKO mice (*Figure 4D–F*).

Finally, we asked if islet hyper-vascularization during development is also responsible for altered expression of genes involved in calcium signaling, insulin granule exocytosis, and β-cell maturation in *Adrb2* cKO islets. Neutralizing VEGFR2 activity at neonatal stages restored the expression of *Cacna1c*, *Pclo*, *Rph3al*, *Snap25*, *Neurod1* and *Npy* in islets isolated from 2 month old *Adrb2* cKO mice (*Figure 4G*). Additionally, immunostaining analyses revealed normal Glut2 and NPY expression in VEGFR2 neutralizing antibody-injected *Adrb2* cKO islets (*Figure 4—figure supplement 1B*). Together, these results support a causal link between Adrb2-mediated control of the intra-islet vasculature via regulating VEGF-A expression during development and the later effects on adult β-cell function and glucose homeostasis.

## Discussion

During development, negative regulatory pathways are as important as positive pathways to counter un-restricted growth of tissues/organs. Here, we reveal a previously uncharacterized role for β2-adrenergic receptors (Adrb2) in acting as an endogenous 'brake' to curtail excessive vascularization during islet development to thereby influence blood glucose homeostasis later in life. Our findings support a scenario (*Figure 5*) where Adrb2, acting in β-cells, suppresses VEGF-A production to limit excessive intra-islet vasculature during islet maturation. Islet endothelial cells are critical for regulation of insulin biosynthesis and exocytosis in neighboring β-cells. Loss of pancreatic Adrb2 disrupts the balance in bi-directional signaling between islet β-cells and endothelial cells, resulting in hyper-vascularized islets and enhanced insulin production. These early effects of Adrb2 on islet vasculature are essential for insulin secretion in adult islets and for glucose homeostasis, in part via transcriptional regulation of key components of calcium signaling, exocytotic machinery, and functional maturation in β-cells. Unexpectedly, we found that the phenotypes in insulin secretion and glucose tolerance upon pancreatic Adrb2 loss are specific to female mice, revealing an Adrb2-mediated sexually dimorphic pathway that underlies islet development and mature function.

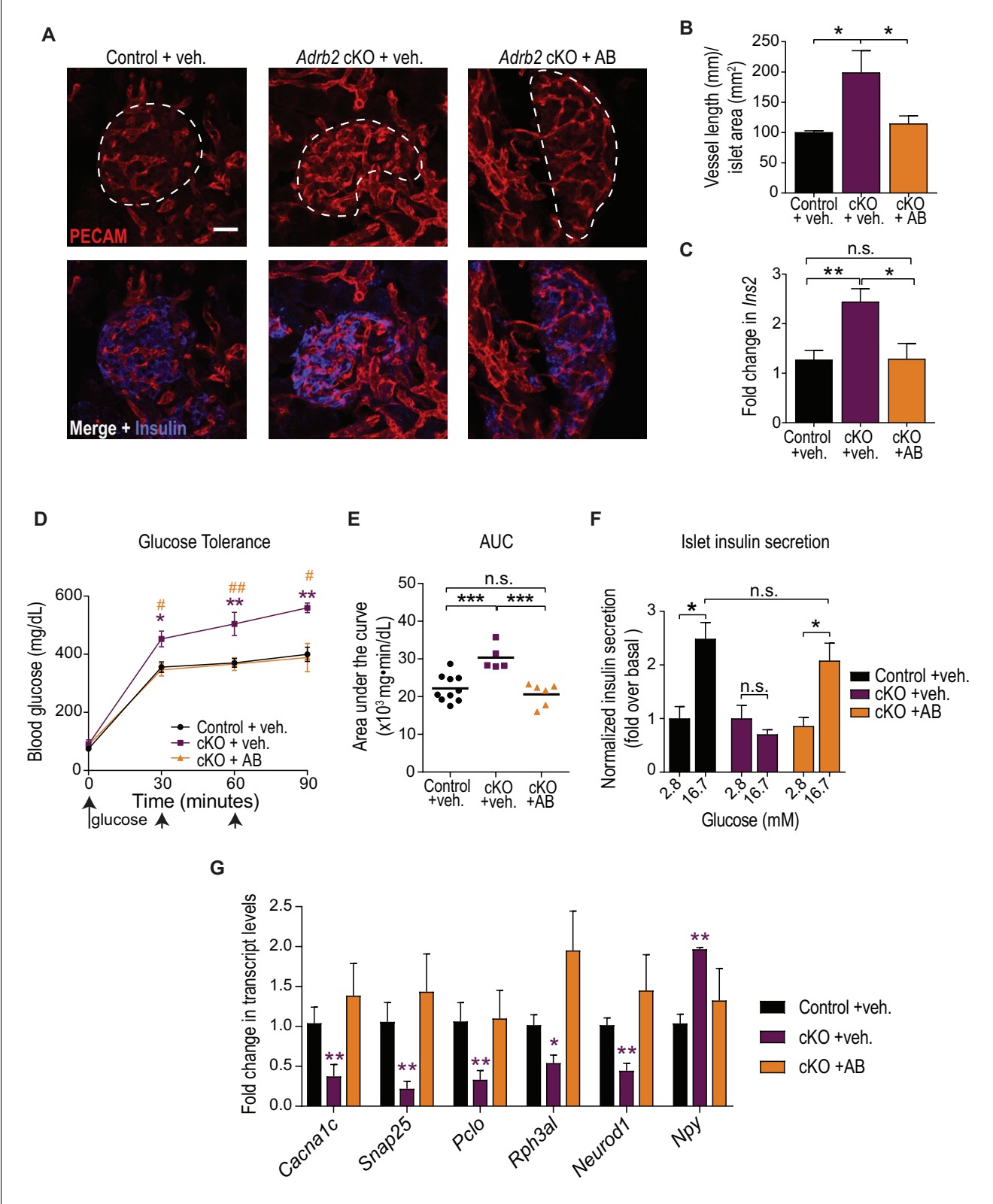

**Figure 4.** Developmental VEGF blockade rescues defects in islet vasculature, insulin secretion, and glucose tolerance in *Adrb2* cKO mice. (**A**) VEGFR2 antibody injections rescue islet hyper-vascularization in neonatal *Adrb2* cKO mice. Islets are outlined in dashed lines. Scale bar, 25 μm. (**B**) Quantification of total vessel length (mm) per islet area (mm²). Means ± SEM from three control and *Adrb2* cKO+ vehicle injected, and 4 *Adrb2* cKO +antibody injected mice. *p<0.05, one-way ANOVA with Tukey's post-test. (**C**) *Ins2* levels are normalized in VEGFR2 antibody-injected *Adrb2* cKO

*Figure 4 continued on next page*

*Figure 4 continued*

neonates. Means ± SEM from 12 control (*Adrb2^{f/f}*)+vehicle injected, 7 *Adrb2* cKO +vehicle injected, and 5 *Adrb2* cKO +antibody injected mice. *p<0.05, **p<0.01, one-way ANOVA with Tukey's post-test. (D) Neonatal administration of VEGFR2 blocking antibody (P0–P6) rescues glucose intolerance in adult female *Adrb2* cKO mice. Means ± SEM for n = 10 control+vehicle, five mutant +vehicle, and six mutant +antibody injected mice. *p<0.05, **p<0.01 *Adrb2* cKO +vehicle significantly different from control *Adrb2^{f/f}* mice, and #p<0.05, ##p<0.01 *Adrb2* cKO +vehicle significantly different from *Adrb2* cKO +antibody injected mice, one-way ANOVA with Tukey's post-test. (E) Area under the curve (AUC) for glucose tolerance. ***p<0.001, n.s. not significant, one-way ANOVA with Tukey's post-test. (F) Rescue of adult islet GSIS in *Adrb2* cKO mice injected with VEGFR2 antibody during the first week of birth. Means ± SEM for n = 4 control+vehicle, three mutant +vehicle, and four mutant +antibody injected mice. *p<0.05, n.s. not significant, two-way ANOVA with Bonferroni's post-test. (G) Developmental VEGFR2 blockade restores expression of genes involved in calcium signaling, exocytosis, and β-cell maturation in adult *Adrb2* cKO mice. Transcript levels were assessed by qRT-PCR analysis and normalized to 18S RNA. Means ± SEM and expressed as fold-change relative to control *Adrb2^{f/f}*+ vehicle values. n = 3 control+vehicle, 4 – 6 mutant +vehicle, and three mutant +antibody injected mice. *p<0.05, **p<0.01, one-sample *t*-test and *t*-test where *Adrb2* cKO values are significantly different from both control and cKO +AB values.

DOI: https://doi.org/10.7554/eLife.39689.015

The following source data and figure supplements are available for figure 4:

**Source data 1.** This spreadsheet includes raw data for islet morphology, metabolic analyses, and gene expression after VEGF-A blockade.
DOI: https://doi.org/10.7554/eLife.39689.018
**Figure supplement 1.** VEGFR2 neutralizing antibody treatment does not affect *Vegfa* levels in *Adrb2* cKO mice, but restores Glut2 and NPY expression.
DOI: https://doi.org/10.7554/eLife.39689.016
**Figure supplement 1—source data 1.** This spreadsheet includes raw data for insulin transcript levels after VEGF-A blockade.
DOI: https://doi.org/10.7554/eLife.39689.017

Islets constitute less than 2% of the total pancreatic volume, but receive disproportionately greater blood supply (*Jansson and Hellerström, 1983*), in part, because of the exquisite dependence of islet β-cells on a high and sustained oxygen supply for aerobic glycolysis. An intimate relationship between β-cells and intra-islet microvasculature is critical for multiple aspects of islet biology including β-cell mass, islet innervation, insulin secretion, and regenerative capacity of islets (*Brissova et al., 2014*; *Brissova et al., 2006*; *Cai et al., 2012*; *Eberhard et al., 2010*; *Hogan and Hull, 2017*; *Reinert et al., 2014*). This cross-talk between islet endothelial and β-cells is under precise regulation (*Brissova et al., 2006*; *Cai et al., 2012*), and a set-point of endothelial to β-cell ratios is critical for mature islet function (*Hogan and Hull, 2017*). In mice, hypo- or hyper-vascularization of newly formed islets by β-cell-specific deletion (*Brissova et al., 2006*) or over-expression of VEGF-A (*Cai et al., 2012*), respectively, both elicit detrimental effects on β-cell mass, insulin secretion, and glucose tolerance, suggesting that VEGF-A levels in β-cells must be maintained within a narrow range. Surprisingly, inactivation of VEGF-A and subsequent islet hypo-vascularization in adult islets did not perturb islet mass and insulin secretion, and elicited only modest elevations in blood glucose (*D'Hoker et al., 2013*; *Reinert et al., 2013*). Together, these studies imply that the influence of the intra-islet vasculature and VEGF-A levels on islet morphology and function is predominantly exerted during development. However, the factors modulating the precise control of β-cell VEGF-A expression, and the link between early islet vascular patterning and later metabolic functions, have remained poorly defined. Here, we reveal that Adrb2 activity in β-cells constitutes an endogenous brake to suppress VEGF-A production and curtail excessive endothelial cell expansion during the first week of birth in mice, a period that is characterized by pronounced growth of intra-islet capillary network and endothelial cell proliferation (*Johansson et al., 2006a*).

The precise molecular mechanisms by which Adrb2 regulates VEGF-A production in β-cells remain to be determined. Adrb2 is activated both by circulating epinephrine, derived from adrenal glands, and norepinephrine, secreted locally by sympathetic nerves, although Adrb2 has 100-fold higher affinity for epinephrine (*Molinoff, 1984*). Our findings that epinephrine, but not norepinephrine, suppressed VEGF-A expression in MIN6 cells, together with observations of differences in phenotypes between *Adrb2* cKO and sympathectomized mice (*Borden et al., 2013*), suggest that epinephrine is the likely ligand responsible for Adrb2-mediated effects on VEGF-A levels and islet vasculature. Adrb2 is a prototypical GPCR that is classically thought to couple with $G\alpha_s$ proteins and activate transcriptional events via a cAMP-PKA-CREB pathway, which stimulates VEGF-A expression in several cell types (*Claffey et al., 1992*; *Lee et al., 2009*; *Wu et al., 2007*). However, association

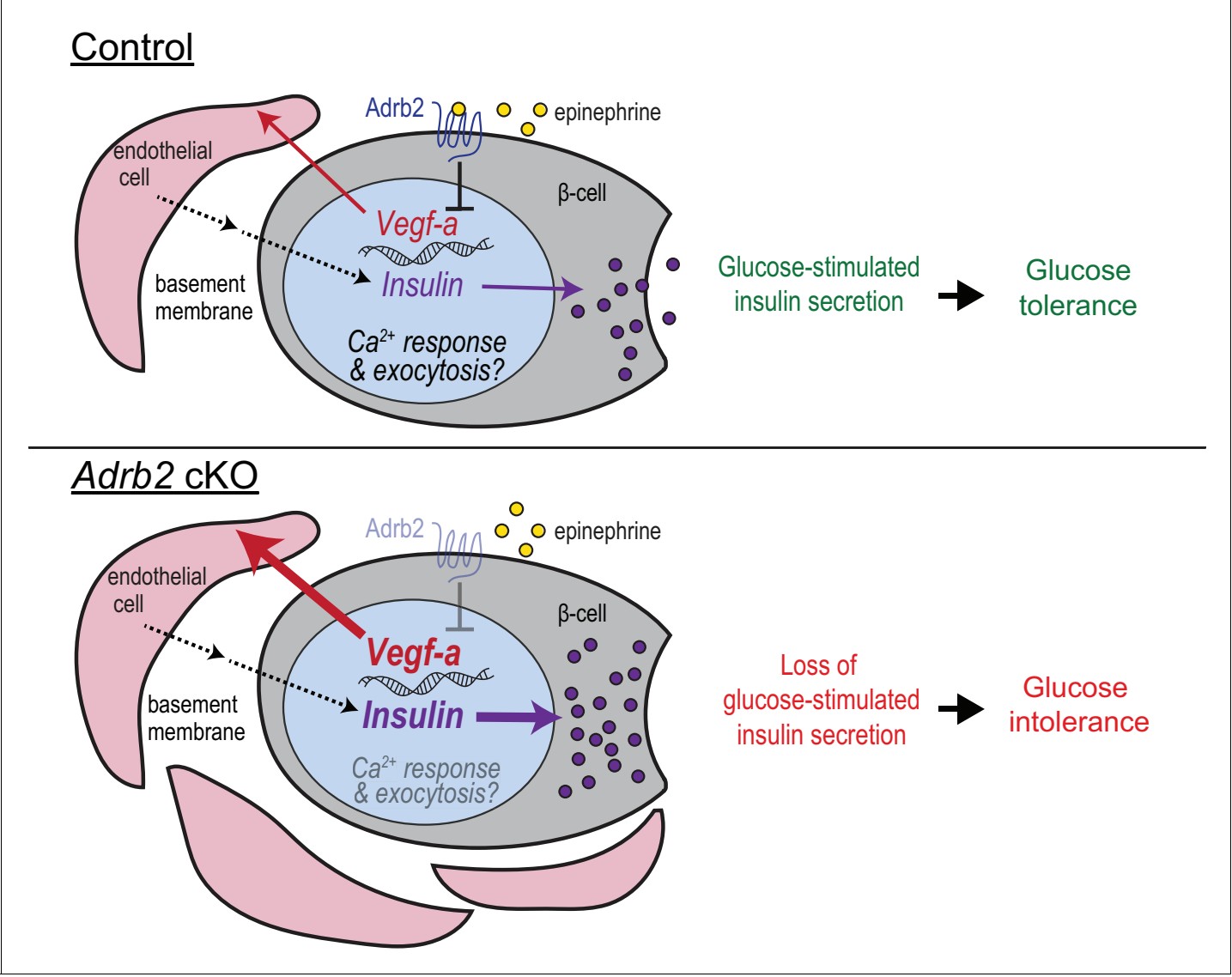

**Figure 5.** Adrb2 regulates adult islet function by controlling bi-directional signaling between β-cells and islet vasculature during development. During islet development, β-cell-Adrb2 receptors suppress VEGF-A expression to limit excessive growth of intra-islet endothelial cells. Endothelial cells, in turn, are critical for precise regulation of insulin gene expression in neighboring β-cells, via producing vascular basement membrane proteins, and likely for transcriptional regulation of key components of insulin exocytosis. Loss of Adrb2 disrupts bi-directional signaling between islet β-cells and endothelial cells during development, resulting in islet hyper-vascularization, aberrant insulin expression, and impaired glucose-stimulated insulin secretion in adult islets.

DOI: https://doi.org/10.7554/eLife.39689.019

of GPCRs with specific Gα proteins is dependent upon the cellular context. Our findings suggest that in pancreatic β-cells, Adrb2 might act via $G\alpha_i$ to negatively regulate cAMP-mediated transcriptional signaling. In support of this idea, loss of Adrb2 enhanced CREB phosphorylation in bone osteoblasts (*Kajimura et al., 2011*).

Both neonatal and adult *Adrb2* cKO islets showed an up-regulation of insulin content, despite a profound impairment in insulin secretion. It is reasonable to assume that impaired insulin secretion in mutant β-cells leads to an accumulation of insulin granules. However, insulin transcript levels were also elevated in *Adrb2* cKO islets, suggesting that enhanced synthesis could also be an important factor contributing to elevated insulin content. Vascular endothelial cells instruct β-cell gene expression and maturation by supplying extracellular matrix molecules and soluble factors including hepatocyte growth factor and thrombospondin-1 (*Eberhard et al., 2010*; *Hogan and Hull, 2017*;

*Johansson et al., 2009*; *Johansson et al., 2006b*; *Olerud et al., 2011*). In particular, intra-islet endothelial cells regulate insulin gene transcription by producing vascular basement membrane proteins (*Kaido et al., 2004*; *Nikolova et al., 2006*; *Nikolova et al., 2007*). Our results that VEGFR2 neutralization restores insulin transcript levels in *Adrb2* cKO islets support the notion that enhanced insulin expression is a consequence of aberrant VEGF-A signaling and perturbed islet vasculature.

We reason that the defect in mature β-cell secretory function in mutant islets is, in part, due to dysregulated expression of genes associated with β-cell maturation, calcium responsiveness, and insulin granule exocytosis. mRNA analyses of β-cell genes in adult *Adrb2* cKO islets revealed decreased *Neurod1* and *Glut2* but enhanced *Npy* expression, which are characteristics of juvenile β-cells (*Gu et al., 2010*), although other late maturation markers such as *Ucn3* (*Blum et al., 2012*; *van der Meulen et al., 2012*) were unchanged. Adrb2 loss also did not elicit enhanced β-cell replication in adult islets, which has been associated with functional immaturity (*Puri et al., 2018*). NPY expression declines postnatally with β-cell maturation, and its re-emergence in animal models of β-cell dysfunction, for example, in β-cell-specific *Neurod1* knockout mice (*Gu et al., 2010*), or during diabetes (*Rodnoi et al., 2017*), is thought to contribute to altered β-cell identity and impaired GSIS. Thus, diminished expression of genes essential for GSIS (*Glut2*) and de-repression of disallowed genes (*Npy*) could alter β-cell maturation and/or identity and eventually contribute to the secretory dysfunction observed in adult *Adrb2* cKO islets. It is unclear if these gene expression changes reflect compensatory efforts to counter the excessive insulin production that develops neonatally and protect against hypoglycemia, or a *bonafide* effect of endothelial cells in instructing mature β-cell phenotype. Developmental VEGFR2 blockade restored the gene expression of β-cell maturation and GSIS machinery in *Adrb2* cKO islets, suggesting that effects of Adrb2 loss are secondary to the disruption of islet endothelial cells. Of note, in contrast to previous studies where VEGF-A over-expression and islet vascular hypertrophy is accompanied by β-cell loss (*Agudo et al., 2012*; *Brissova et al., 2014*; *Cai et al., 2012*), we did not observe major disruptions in β-cell proliferation, islet size, or architecture in *Adrb2* cKO animals. This could reflect differences in the timing and levels of VEGF-A over-expression elicited by Adrb2 loss versus transgenic over-expression of VEGF-A. Early increases in VEGF-A expression and islet vascular density preceding the onset of hyperglycemia have been observed in animal models of type two diabetes (*Li et al., 2006*). An expanded islet capillary network has also been noted in human islets from individuals with type two diabetes (*Brissova et al., 2015*), although the patterning and density of the human islet micro-vasculature differs significantly from that in rodents (*Brissova et al., 2015*). Together, these studies underscore the need to fully delineate the pathways underlying the tight control of β-cell-endothelial cell communication and their long-term impact on islet function.

We demonstrate that the effects of pancreatic Adrb2 receptors on glucose homeostasis are predominantly due to its functions during a critical window during neonatal development. Islet Adrb2 expression declines markedly during postnatal stages with adult islets (P60) showing significantly depleted expression relative to neonatal stages (P2 and P6). Consistent with this expression pattern, Adrb2 deletion from neonatal, but not adult β-cells, impaired glucose tolerance and insulin secretion. That Adrb2 loss from adult murine β-cells did not impair insulin secretion is counter to previously reported effects of β2-adrenergic agonists on acutely stimulating insulin secretion in adult human islets (*Lacey et al., 1993*). Thus, either the effects of Adrb2 agonists on adult human islets are indirect, perhaps via regulating glucagon release from α-cells (*Ahrén and Scherstén, 1986*), or there are species-specific differences in adult Adrb2 functions in β-cells. β-adrenergic receptors have been best characterized for mediating diverse physiological responses to stress in adult mammals, and are one of the most common targets of therapeutic drugs. β-adrenergic receptor antagonists (β-blockers) are commonly used to treat cardiovascular disease and hypertension, including in pregnant women (*Easterling, 2014*; *Parati and Esler, 2012*; *Sharma et al., 2001*), and administered to infants for treating hemangiomas (*Chinnadurai et al., 2016*). Our findings in mice suggest that inhibition of pancreatic Adrb2 activity during development could have a lasting impact on glycemic control.

Unexpectedly, we found that loss of Adrb2 specifically elicits metabolic phenotypes in female mice. Traditionally, the majority of rodent studies on insulin secretion and glucose homeostasis have employed adult males (*Gannon et al., 2018*), resulting in a scarcity of information on sex differences in β-cell development and function. However, it has been noted in different inbred mouse strains, that males are generally less glucose tolerant than females (*Gannon et al., 2018*; *Goren et al.,*

*2004*; *Lavine et al., 1971*), likely, in part, due to lower glucose-stimulated insulin secretion (*Gannon et al., 2018*; *Goren et al., 2004*). Similarly, clinical studies in humans support that healthy women tend to have increased insulin secretion after a meal or in response to an oral glucose load compared to men (*Basu et al., 2017*; *Basu et al., 2006*; *Horie et al., 2018*). Furthermore, isolated female islets show greater glucose-stimulated insulin secretion compared to male islets (*Hall et al., 2014*). Sex differences in islet function could arise from one or a combination of multiple factors including, but not limited to, differences in sex chromosomes, developmental effects of steroid sex hormones, in particular testosterone, which surges perinatally and exerts major organizational changes on neuronal circuits, or activity of gonadal hormones following puberty (*Gannon et al., 2018*). The molecular underpinnings of sexually dimorphic Adrb2 functions in islets remain to be defined. However, we and others (*Berger et al., 2015*) have observed that islet Adrb2 expression is sexually dimorphic with expression in male islets being lower than females. During pregnancy, β-cell mass and intra-islet vasculature undergo dramatic expansions to accommodate the needs of the growing fetus (*Johansson et al., 2006b*; *Mauvais-Jarvis, 2015*; *Sorenson and Brelje, 1997*). Negative regulatory mechanisms, such as Adrb2-mediated restriction of vascular growth and insulin production, may be particularly critical for adaptive structural and functional changes in female islets to meet the metabolic demands of pregnancy and lactation. Although Adrb2 deletion in adult females does not affect β-cell function, whether this pathway is relevant under specific circumstances such as pregnancy remains to be determined.

# Materials and methods

## Key resources table

| Reagent type (species) or resource | Designation | Source or reference | Identifiers | Additional information |
|---|---|---|---|---|
| Genetic reagent (*M.musculus*) | B6.FVB-Tg(Pdx1-cre)6Tuv/Nci; Pdx1-Cre | NCI Frederick mouse repository | RRID: IMSR_NCIMR:01XL5 | |
| Genetic reagent (*M.musculus*) | B6.Cg-Tg(Ins1-EGFP)1Hara/J; MIP-GFP | The Jackson Laboratory | RRID: IMSR_JAX:006864 | |
| Genetic reagent (*M. musculus*) | *Adrb2^{f/f}* | *Hinoi et al. (2008)* | | Dr. Gerard Karsenty (Columbia University) |
| Genetic reagent (*M. musculus*) | Tg(Pdx1-cre/Esr1*)#Dam/J; *Pdx1-cre/Esr1*^{*Dam}* | The Jackson Laboratory | RRID: IMSR_JAX:024968 | |
| Genetic reagent (*M. musculus*) | *C57BL/6J* | The Jackson Laboratory | RRID: IMSR_JAX:000664 | |
| Cell line (*M. musculus*) | MIN6 | | | Dr. Jun-ichi Miyazaki (Osaka University) |
| Antibody | Guinea Pig anti-insulin | Dako | Cat # A0564; RRID: AB_10013624 | IHC (1:300) |
| Antibody | Mouse anti-glucagon | Abcam | Cat # ab10988; RRID: AB_297642 | IHC (1:500) |
| Antibody | Rat anti-PECAM-1/CD-31 | BD Biosciences | Cat # 550274; RRID: AB_393571 | IHC (1:300) |
| Antibody | Rabbit anti-collagen IV | Abcam | Cat # ab19808; RRID: AB_445160 | IHC (1:500) |
| Antibody | Chicken anti-laminin | LifeSpan Biosciences | Cat # LS-C96142; RRID: AB_2134058 | IHC (1:200) |
| Antibody | Mouse VEGFR2/Kdr/Flk-1 | R and D Systems | Cat # AF644; RRID: AB_355500 | 1 µg injection dose |
| Antibody | Rabbit anti-NPY | BMA Biomedicals | Cat # T-4070; RRID: AB_518504 | IHC (1:1000) |
| Antibody | Rabbit anti-Glut2 | Alpha Diagnostics | Cat # GT21-A; RRID: AB_1616640 | IHC (1:300) |

*Continued on next page*

*Continued*

| Reagent type (species) or resource | Designation | Source or reference | Identifiers | Additional information |
|---|---|---|---|---|
| Sequence-based reagent | Adra1a TaqMan Probe | ThermoFisher | Assay ID: Mm00442668_m1 | |
| Sequence-based reagent | Adra1b TaqMan Probe | ThermoFisher | Assay ID: Mm00431685_m1 | |
| Sequence-based reagent | Adra1d TaqMan Probe | ThermoFisher | Assay ID: Mm01328600_m1 | |
| Sequence-based reagent | Adra2a TaqMan Probe | ThermoFisher | Assay ID: Mm00845383_s1 | |
| Sequence-based reagent | Adra2b TaqMan Probe | ThermoFisher | Assay ID: Mm00477390_s1 | |
| Sequence-based reagent | Adra2c TaqMan Probe | ThermoFisher | Assay ID: Mm00431686_s1 | |
| Sequence-based reagent | Adrb1 TaqMan Probe | ThermoFisher | Assay ID: Mm00431701_s1 | |
| Sequence-based reagent | Adrb2 TaqMan Probe | ThermoFisher | Assay ID: Mm02524224_s1 | |
| Sequence-based reagent | Adrb3 TaqMan Probe | ThermoFisher | Assay ID: Mm02601819_g1 | |
| Sequence-based reagent | Eukaryotic 18S rRNA Endogenous Control (VIC/MGB probe, primer limited) | ThermoFisher | Cat # 4319413E | |
| Commercial Assay | Ultrasensitive mouse insulin ELISA kit | Crystal Chem | Cat # 90080 | |
| Chemical compound, drug | Fluo-4, AM, cell permeant | ThermoFisher | Cat # F14201 | |

## Animals and cell lines

All procedures relating to animal care and treatment conformed to The Johns Hopkins University Animal Care and Use Committee (ACUC) and NIH guidelines. Animals were group housed in a standard 12:12 light-dark cycle. Since phenotypes were specific to female mutant mice, animals of both sexes were used for initial analyses in *Figure 1*, but only females were used for later studies described in *Figures 2–4*, unless otherwise noted. The following mouse lines were used in this study; *B6.FVB-Tg(Pdx1-cre)6Tuv/Nci* (NCI Frederick Mouse Repository, strain 01XL5), *B6.Cg-Tg(Ins1-EGFP) 1Hara/J* (Jackson Laboratory, stock no. 006864), *Adrb2f/f* (*Hinoi et al., 2008*), *Tg(Pdx1-cre/Esr1*) #Dam/J* (Jackson Laboratory, stock no. 024968), and *C57BL/6J* (Jackson Laboratory, stock no. 000664). MIN6 cells (sex of animal of origin unknown) were obtained from Dr. Jun-ichi Miyazaki (Osaka University). MIN6 cells were used between passages 8 – 22 when they have been reported to exhibit intact GSIS (*Cheng et al., 2012*; *Miyazaki et al., 1990*). We confirmed insulin and Glut2 immunoreactivity, and routinely examined cellular morphology. Mycoplasma testing of MIN6 cells was negative based on PCR profiling of mycoplasmic 16S rRNA (*Drexler and Uphoff, 2002*). Cells were maintained in high glucose DMEM +L glutamine supplemented with 10% fetal bovine serum (FBS), 5 mL sodium pyruvate, 5 U/L penicillin-streptomycin, and 2 μl β-mercaptoethanol. Incubation conditions were humidified 37°C and 5% CO2.

## Metabolic assays
### Glucose tolerance

Mice were individually housed and fasted overnight (16 hr) before being given a 2 g/kg intraperitoneal (IP) glucose injection. Blood glucose levels were measured with a OneTouch Ultra two glucometer before glucose injection, and 2 additional 2 g/kg glucose injections were given at 30 min and 60 min after the first injection. Glucose levels were monitored over a 1.5 hr period. Area Under the Curve (AUC) was determined for each animal using GraphPad Prism 7.

## In vivo insulin secretion

Mice were individually housed and fasted overnight (16 hr) before being given a 3 g/kg IP glucose injection. Plasma insulin levels were measured with an Ultrasensitive Insulin ELISA kit (Crystal Chem, Elk Grove Village, IL) before glucose injection and over a 1 hr period following injection.

## In vivo insulin sensitivity

Mice were individually housed overnight with *ad libitum* food and water before being given a 0.75 U/kg IP insulin injection (Novolin-R, Novo Nordisk, Princeton, NJ). Blood glucose levels were measured with a OneTouch Ultra two glucometer before injection, and every 15 min post-injection over a 1 hr period.

## Mouse islet isolations

Islets were isolated from neonatal or 2 – 4 month-old mice as previously described (*Wollheim et al., 1990*). Briefly, islets were isolated by collagenase distension through the bile duct for adults or by injection of dissected pancreata for neonates (Collagenase P [Roche, Basel, Switzerland], 0.375 – 0.4 mg/mL in HBSS) and digestion at 37°C. Digested pancreata were washed with HBSS +0.1% BSA and subjected to discontinuous density gradient using histopaque (6:5 Histopaque 1119:1077; Sigma, St. Louis, MO). The islet layer (found at the interface) was collected and islets were handpicked under an inverted microscope for subsequent analysis.

## In vitro insulin secretion

Isolated islets were cultured overnight in RPMI-1640 (5% FBS, 5 U/L penicillin-streptomycin). Islets were then handpicked to be size-matched, and 5 – 10 islets/mouse were washed and pre-incubated for 1 hr in Krebs-Ringer HEPES buffer (KRHB) containing 2.8 mM glucose. Islets were then incubated in 2.8 mM or 16.7 mM glucose, or 30 mM KCl in KRHB for another 30 min. Supernatant fractions were removed, and islets lysed in acid-ethanol overnight and subsequently neutralized in Tris buffer (0.885M). Both cellular and supernatant fractions were subjected to insulin ELISA (Crystal Chem).

## Calcium imaging

Isolated islets were cultured overnight on laminin-coated (0.05 – 0.1 mg/mL in HBSS + $Ca^{2+}$ + $Mg^{2+}$) MatTek 35 mm, No. 1.5 coverslip, 20 mm glass diameter dishes in high glucose DMEM+L glutamine supplemented with 10% fetal bovine serum (FBS), 5 mL sodium pyruvate, 5 U/L penicillin-streptomycin, and 2 μl β-mercaptoethanol. Adhered islets were loaded with 4 μM Fluo-4 AM (ThermoFisher, Waltham, MA) for 45 min in KRHB with 2.8 mM glucose, then washed and incubated for 30 additional minutes in KRHB with 2.8 mM glucose on a stage incubator (37°C and 5% $CO_2$) for equilibration prior to imaging. Ten z-stacks representing 1 μm optical slices of individual islets were collected every 15 s on a Zeiss AxioObserver Yokogawa CSU-X1 spinning disk confocal (40 X objective) equipped with dual Evolve EMCCDs and 405, 488, 555, and 633 nm lasers. Zen Blue (2012) image collection software was used to maintain experimental parameters between individual experiments. Concentrated glucose was added manually after 5 min of imaging (to accomplish a final concentration 20 mM glucose) and imaging continued for 10 min, before addition of concentrated KCl (final concentration 30 mM). Imaging was continued for 10 additional minutes, for a total of 25 min. No more than 30 s elapsed between solution changes. Analysis of calcium flux over time was measured using FIJI. Maximum intensity projections were generated for each time point, and the Region Of Interest (ROI) manager was used to manually outline 10 cells per animal. For each time point, the Fluo-4 fluorescence per islet area was calculated using integrated density (FIJI). Only cells with visible basal fluorescence in low glucose were chosen for quantification purposes. The fluorescence intensity for each time point was then compared to the average basal (low glucose) fluorescence intensity. The average intensity of the 10 cells per animal was plotted over time using GraphPad Prism 7, and the area under the curve (AUC) was determined for high glucose and KCl conditions. The AUCs for control and *Adrb2* cKO islet cells were compared using a two-tailed *t*-test.

## Tamoxifen injections

For adult deletion, beginning at 5 – 6 weeks of age, *Adrb2* i-cKO mice were injected subcutaneously with 180 mg/kg tamoxifen (Sigma) in corn oil or corn oil alone, every day for 5 days. 4 weeks after

the last injection, vehicle- and tamoxifen-injected mice were subjected to metabolic assays. For neonatal deletion, *Adrb2* i-cKO mice were injected subcutaneously with 180 mg/kg tamoxifen in corn oil or corn oil alone at P0 and P1. Injection sites were sealed with VetBond Tissue Adhesive (3M, Maplewood, MN). Glucose tolerance and insulin secretion assays were done when the mice were 2 months old.

## Immunohistochemistry

Pancreata were dissected, fixed in 4% paraformaldehyde in PBS (PFA, Sigma), cryo-protected in 30% sucrose in PBS (Sigma), equilibrated in 1:1 30% sucrose:OCT (Sakura Finetek, Torrance, CA), and embedded in OCT before being cryo-sectioned. Sections were frozen at −80°C for storage. 50 μm cryosections were taken from P6 mouse pancreata, then washed with PBS, permeabilized in 1% Triton X-100 in PBS, and blocked for 1 hr at room temperature using 5% goat serum (GS) in PBS + 0.1% Triton X-100. Sections were then incubated for 1 – 2 nights at 4°C with either: rat anti-CD31 (PECAM1; BD Biosciences, San Jose, CA, cat# 550274) antibody (1:300), rabbit anti-collagen IV (1:500; Abcam, Cambridge, UK cat# ab19808), chicken anti-laminin (1:200; LifeSpan Biosciences, Seattle, WA cat# LS-C96142), or rabbit anti-NPY (1:1000; BMA Biomedicals, Augst, Switzerland, cat#T-4070) in conjunction with guinea pig anti-insulin (1:300; Dako, Agilent, Santa Clara, CA, cat# A0564). For Glut-2 immunostaining, sections were first treated with antigen retrieval solution (10 mM sodium citrate, 0.05% Tween-20, pH 6) at 95°C for 10 min followed by a 0.1M glycine wash, blocking, and incubation with rabbit anti-Glut2 (1:300; Alpha Diagnostics, Owings Mills, MD, cat#GT21-A). Following PBS washes, sections were then incubated with anti-rat Alexa-546, anti-rabbit Alexa 488 or 647, anti-chicken 488 or 647, and anti-guinea pig 488, 546, or 647 secondary antibodies (1:200; ThermoFisher). Sections were then washed in PBS and mounted in Fluoromount Aqueous Mounting Medium containing 100 μg/mL DAPI. Z-stacks of 1 μm optical slices were taken using a Zeiss LSM 700 confocal microscope equipped with 405, 488, 555, and 633 lasers. Maximum intensity projections were generated with FIJI.

Quantification of total vessel length per islet area was determined from multiple islets per animal, and at least three independent animals using Simple Neurite Tracer (FIJI plugin).

## Islet morphometric analyses

Cryosections were taken from P6 mouse pancreata at 10 μm, every 200 μm throughout the whole pancreas. Sections were washed in PBS, permeabilized in 1% Triton X-100 in PBS, and blocked for 1 hr at room temperature using 5% goat serum (GS) in PBS + 0.1% Triton X-100. Sections were then incubated overnight at 4°C with a guinea pig anti-insulin antibody (1:300) and a mouse IgG1 anti-glucagon antibody (1:500; Abcam cat# ab10988). Following PBS washes, sections were then incubated with anti-guinea pig Alexa-546 and anti-mouse IgG1 Alexa 488 secondary antibodies (1:200; ThermoFisher). Sections were then washed in PBS and mounted in Fluoromount Aqueous Mounting Medium containing 100 μg/mL DAPI. Every islet in the collected sections was imaged on a Zeiss Axio-Imager. The number of α- and β-cells was counted in each islet in each section, and the total α- and β-cell number counted was divided by the number of sections to determine endocrine cell number.

## EdU labeling

*Adrb2* cKO and control litter-mate mice were injected intraperitoneally with 100 μg of EdU (dissolved in 3:1 PBS:DMSO) 24 hr before harvesting tissues. Pancreata were collected and processed for immunohistochemistry as described above. 12 μm thick cryo-sections were collected and stained for insulin and anti-guinea pig 546, then processed for 30 min at room temperature in EdU reaction cocktail (ThermoFisher EdU kit C10337; Click-iT buffer, Buffer additive, CuSO4 solution, and Alexa-Fluor 488). Sections were then washed in PBS + 0.1% TritonX-100 and mounted with VectaShield +DAPI. Images were collected using a Zeiss LSM 700 confocal microscope (405, 488, and 555 nm lasers). Islets from at least four sections were quantified for insulin and EdU/Insulin double-positive cells to assess β-cell proliferation.

## Transmission electron microscopy

Pancreata were dissected and cut into ~1 mm³ pieces, then fixed in 3% formaldehyde +1.5% glutaraldehyde+2.5% sucrose in 0.1M NaCacodylate +5 mM $CaCl_2$, pH 7.4 at room temperature for 1 hr. Pieces were then washed in 2.5% sucrose in 0.1M NaCacodylate +5 mM $CaCl_2$ and post-fixed with 1% Palade's $OsO_4$ for 1 hr on ice, followed by incubation in Kellenberger's uranyl acetate overnight at room temperature. Samples were then dehydrated with a graded alcohol series (50%, 75%, 95%, and 100% ethanol, then propylene oxide), embedded in Epon, and sectioned (~90 nm) and collected onto EM grids. Grids were then imaged using an FEI Tecnai-12 TWIN transmission electron microscope operating at 100 kV and a SIS MegaView III wide-angle camera. For adult granule docking analyses, granules within 50 nm of a plasma membrane facing a lumen were considered docked.

## Cell sorting

Pancreata from P6 control and *Adrb2* cKO animals in a MIP-GFP background (*Hara et al., 2003*) were digested in 0.05% trypsin, 0.53M EDTA for 15 min at 37°C with intermittent trituration. Samples were then neutralized with Hank's Balanced Salt Solution (HBSS, Gibco Gaithersburg, MD)+2% FBS, 5 mM EDTA, and 10 µg/mL DNAse and filtered (70 µm). Dissociated cells were then sorted using a Sony Biotechnology SH 800 cell sorter, and GFP-positive cells separated based on detection using the FL1 channel (absorption 530 nm).

## MIN6 cell treatments

MIN6 cells less than passage 30 were plated in 6-well dishes and allowed to grow to ~90% confluency. Cells were then treated with either control (water or methanol) or 10 µM Adrb2 agonist for 16 hr before being collected for qPCR. Norepinephrine (L-(−)-Norepinephrine (+)-bitartrate salt monohydrate, Sigma cat# A9512) and epinephrine (±)-Epinephrine hydrochloride, Sigma cat# E4642) were dissolved in water; salbutamol (Sigma cat# S8260) was dissolved in methanol.

## qRT-PCR

RNA from either whole pancreata (P0, P2, P6), isolated islets (2 – 4 month-old mice), or MIN6 cells was isolated using TRIzol RNA extraction reagent phenol-chloroform extraction and ethanol precipitation. RNA from sorted cells was isolated using Direct-zol RNA Microprep (Zymo Irvine, CA) kit. 1.5 – 2 µg of total RNA was reverse-transcribed into cDNA with M-MLV Reverse Transcriptase (Promega Madison, WI). TaqMan probes (ThermoFisher) were used to quantitatively determine transcript levels of the adrenergic receptors, while Maxima SYBR Green/Rox Q-PCR Master Mix (ThermoFisher) reagents and primer sets (listed in *Table 1*) were used to determine transcript levels for other genes. Both assays were performed in a StepOnePlus Real-Time PCR System (ThermoFisher). Fold change in transcript levels was calculated using the $2^{(-\Delta\Delta Ct)}$ method, and 18S was used as an endogenous control for normalization.

## VEGFR2 antibody injections

*Adrb2* cKO pups were injected with either 1 µg of VEGFR2/Flk1 blocking antibody (cat#AF644, R and D Minneapolis, MN) or vehicle at P0 and P3, then either collected at P6 for RNA isolation or immunohistochemistry, or given a third injection and allowed to grow to adulthood. Pups given three injections were subjected to glucose tolerance tests at 2 months of age, then collected for in vitro islet insulin secretion assays or RNA isolation.

## Quantification and statistical analyses

Sample sizes were similar to those reported in previous publications (*Borden et al., 2013*; *Houtz et al., 2016*). For practical reasons, analyses of endocrine cell counts and quantifications from immunohistochemistry (TH density and vessel length) and ultrastructural analyses (caveolae and fenestrae density, insulin granule density, insulin granule docking) were done in a semi-blinded manner, such that the investigator was aware of the genotypes prior to the experiment, but conducted the staining and data analyses without knowing the genotypes of each sample. All Student's *t*-tests were performed assuming Gaussian distribution, two-tailed, unpaired, and with a confidence interval of 95%, with the exception of fold change analyses (such as for qPCR), which were done using a one-sample *t*-test. One-way ANOVA analyses with post hoc Tukey test were performed when more

**Table 1.** Primer sets used for qPCR.

| Sequence | Source | Identifier |
|---|---|---|
| Ins2-F:<br>5'-TGTCAAGCAGCACCTTTGTG-3' | This paper | |
| Ins2-R:<br>5'-ACATGGGTGTGTAGAAGAAGCC-3' | This paper | |
| Vegfa-F:<br>5'-GCACATAGAGAGAATGAGCTTCC-3' | Ford et al, 2011 | |
| Vegfa-R:<br>5'-CTCCGCTCTGAACAAGGCT-3' | Ford et al, 2011 | |
| Aldob-F:<br>5'- AGAAGGACAGCCAGGGAAAT-3' | *Gu et al. (2010)* | |
| Aldob-R:<br>5'- GTTCAGAGAGGCCATCAAGC-3' | *Gu et al. (2010)* | |
| PFKL-F:<br>5'- GCTGCAATGGAGAGTTGTGA-3' | *Gu et al. (2010)* | |
| PFKL-R:<br>5'- GGATGTTGAAAGGGTCCTCA-3' | *Gu et al. (2010)* | |
| LDHA-F:<br>5'- TGTCTCCAGCAAAGACTACTGT-3' | *Gu et al. (2010)* | |
| LDHA-R:<br>5'- GACTGTACTTGACAATGTTGGGA-3' | *Gu et al. (2010)* | |
| TPI-F:<br>5'- CCAGGAAGTTCTTCGTTGGGG-3' | *Gu et al. (2010)* | |
| TPI-R:<br>5'- CAAAGTCGATGTAAGCGGTGG-3' | *Gu et al. (2010)* | |
| PKLR-F:<br>5'- TCAAGGCAGGGATGAACATTG-3' | *Gu et al. (2010)* | |
| PKLR-R:<br>5'- CACGGGTCTGTAGCTGAGTG-3' | *Gu et al. (2010)* | |
| Kcnj11-F:<br>5'- AGGGCATTATCCCTGAGGAA −3' | Harvard Primer Bank | PrimerBank ID 6754426a1 |
| Kcnj11-R:<br>5'- TTGCCTTTCTTGGACACGAAG-3' | Harvard Primer Bank | PrimerBank ID 6754426a1 |
| Abcc8-F:<br>5'- TCAACTTGTCTGGTGGTCAGC-3' | *Gu et al. (2010)* | |
| Abcc8-R:<br>5'- GAGCTGAGAAAGGGTCATCCA-3' | *Gu et al. (2010)* | |
| Cacna1c-F:<br>5'-ATGAAAACACGAGGATGTACGTT-3' | Harvard Primer Bank | PrimerBank ID 6753228a1 |
| Cacna1c-R:<br>5'-ACTGACGGTAGAGATGGTTGC-3' | Harvard Primer Bank | PrimerBank ID 6753228a1 |
| Cacna1d-F:<br>5'- AGAGGACCATGCGAACGAG-3' | Harvard Primer Bank | PrimerBank ID 20338999a1 |
| Cacna1d-R:<br>5'- CCTTCACCAGAAATAGGGAGTCT-3' | Harvard Primer Bank | PrimerBank ID 20338999a1 |
| Snap25-F:<br>5'- CAACTGGAACGCATTGAGGAA-3' | Harvard Primer Bank | PrimerBank ID 6755588a1 |
| Snap25-R:<br>5'- GGCCACTACTCCATCCTGATTAT-3' | Harvard Primer Bank | PrimerBank ID 6755588a1 |
| Stx1A-F:<br>5'- AGAGATCCGGGGCTTTATTGA-3' | Harvard Primer Bank | PrimerBank ID 15011853a1 |
| Stx1A-R:<br>5'- AATGCTCTTTAGCTTGGAGCG-3' | Harvard Primer Bank | PrimerBank ID 15011853a1 |
| Pclo-F:<br>5'-TACTCGGACCCATTTGTGAA-3' | *Gu et al. (2010)* | |

*Table 1 continued on next page*

*Table 1 continued*

| Sequence | Source | Identifier |
|---|---|---|
| Pclo-R:<br>5'-TACTGTTTGATTCCACTCGGGATT-3' | *Gu et al. (2010)* | |
| Rph3al-F:<br>5'-GCAGTGGAAATGATCAGTGG-3' | *Gu et al. (2010)* | |
| Rph3al-R:<br>5'-TCAGGCACTGGCTCCTCCTC-3' | *Gu et al. (2010)* | |
| Neurod1-F:<br>5'- ATGACCAAATCATACAGCGAGAG-3' | Harvard Primer Bank | PrimerBank ID 33563268a1 |
| Neurod1-R:<br>5'- TCTGCCTCGTGTTCCTCGT-3' | Harvard Primer Bank | PrimerBank ID 33563268a1 |
| Nkx6.1-F:<br>5'-CTGCACAGTATGGCCGAGATG-3' | Harvard Primer Bank | PrimerBank ID 21450629a1 |
| Nkx6.1-R:<br>5'CCGGGTTATGTGAGCCCAA-3' | Harvard Primer Bank | PrimerBank ID 21450629a1 |
| Npy-F:<br>5'-AGAGATCCAGCCCTGAGACA-3' | *Gu et al. (2010)* | |
| Npy-R:<br>5'-GATGAGGGTGGAAACTTGGA-3' | *Gu et al. (2010)* | |
| Pdx1-F:<br>5'-CCCCAGTTTACAAGCTCGCT-3' | Harvard Primer Bank | PrimerBank ID 22122647a1 |
| Pdx1-R:<br>5'-CTCGGTTCCATTCGGGAAAGG-3' | Harvard Primer Bank | PrimerBank ID 22122647a1 |
| Ucn3-F:<br>5'-AAGCCTCTCCCACAAGTTCTA-3' | Harvard Primer Bank | PrimerBank ID 21492632a1 |
| Ucn3-R:<br>5'-GAGGTGCGTTTGGTTGTCATC-3' | Harvard Primer Bank | PrimerBank ID 21492632a1 |
| Slc2a2-F:<br>5'-TCAGAAGACAAGATCACCGGA-3' | Harvard Primer Bank | PrimerBank ID 13654262a1 |
| Slc2a2-R:<br>5'-GCTGGTGTGACTGTAAGTGGG-3' | Harvard Primer Bank | PrimerBank ID 1 3654262a1 |
| 18S rRNA-F:<br>5'-CGCCGCTAGAGGTGAAATTC-3' | Park et al, 2016 | |
| 18S rRNA-R:<br>5'-TTGGCAAATGCTTTCGCTC-3' | Park et al, 2016 | |

DOI: https://doi.org/10.7554/eLife.39689.020

than two groups were compared. Two-way ANOVA analyses with post hoc Bonferroni's test were performed when there were two independent variables, for example genotype and glucose concentrations. Statistical analyses were based on at least three independent experiments and described in the figure legends. All error bars represent the standard error of the mean (S.E.M.).

## Acknowledgements

We thank H Zhao, M Parsons, M Van Doren, and S Hattar for suggestions on the project and manuscript, members of the Kuruvilla laboratory for discussions, G Karsenty (Columbia) for providing *Adrb2^{f/f}* mice, and Integrated Imaging Center (JHU) for help with imaging. This work was supported by a NIH grant (DK108267) to R.K., NRSA post-doctoral fellowship (F32DK116482) to EEL, and AMC is supported by a NIH training grant (T32GM007231).

## Additional information

### Funding

| Funder | Grant reference number | Author |
|---|---|---|
| National Institutes of Health | R01DK108267 | Rejji Kuruvilla |

The funders had no role in study design, data collection and interpretation, or the decision to submit the work for publication.

### Author contributions

Alexis M Ceasrine, Conceptualization, Data curation, Formal analysis, Validation, Investigation, Visualization, Methodology, Writing—original draft, Writing—review and editing; Eugene E Lin, Data curation, Formal analysis, Funding acquisition, Investigation, Visualization, Methodology, Writing—original draft; David N Lumelsky, Radhika Iyer, Data curation, Formal analysis; Rejji Kuruvilla, Conceptualization, Supervision, Funding acquisition, Writing—original draft, Project administration

### Author ORCIDs

Alexis M Ceasrine (iD) http://orcid.org/0000-0003-0000-0513

Rejji Kuruvilla (iD) http://orcid.org/0000-0002-2851-675X

### Ethics

Animal experimentation: All procedures relating to animal care and treatment conformed to The Johns Hopkins University Animal Care and Use Committee (ACUC) and NIH guidelines. All of the animals were handled according to approved institutional ACUC protocols (#MO17A14) of Johns Hopkins University

### Decision letter and Author response

Decision letter https://doi.org/10.7554/eLife.39689.023

Author response https://doi.org/10.7554/eLife.39689.024

## Additional files

### Supplementary files

• Transparent reporting form

DOI: https://doi.org/10.7554/eLife.39689.021

### Data availability

All data generated or analyzed are included in the manuscript and supporting files

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
