## [Decision Letter]

Thank you for submitting your article "*Adrb2* controls glucose homeostasis by developmental regulation of pancreatic islet vasculature" for consideration by *eLife*. Your article has been reviewed by three peer reviewers and the evaluation has been overseen by a Reviewing Editor and Didier Stainier as the Senior Editor. The reviewers have opted to remain anonymous.

The reviewers have discussed the reviews with one another and the Reviewing Editor has drafted this decision to help you prepare a revised submission.

Summary:

In this manuscript, Ceasrine and colleagues analyze the requirement of *Adrb2* for β-cell function and glucose regulation. The manuscript describes the interesting observation that the β2-adrenergic receptor is important for normal β-cell function, but only during a critical post-natal window, and only in female mice. For this study, they employ mouse tools for constitutive and induced deletion of *Adrb2* in pancreatic epithelial cells and β-cells. Overall, the authors make a convincing case that the lack of *Adrb2* leads to excess β-cell *Vegfa* expression during a critical early post-natal window, driving excessive vascularization which then impairs insulin secretion. The rescue experiment that involves transiently blocking *Vegfa* action elegantly demonstrates the latter point. They also establish a feedback loop where β-adrenergic agonists inhibit β-cell *Vegfa* secretion to curb excess islet vascularization.

The paper is well written and the data are of high quality and uniformly well presented. With the exception of a number of (relatively minor) issues that are straightforward to address, the conclusions are supported by the data presented. The major concerns are predominantly related to the organization of the manuscript and a couple of important pieces data that are needed to clarify the sex-specific bias and lack of phenotype in the adult specific β-cell knockout.

Essential revisions:

The organization of the manuscript is difficult to follow. The authors present some of the cKO data first, followed by the i-cKO data, and then back to the cKO data. It would make more sense to combine the cKO data. And more importantly, it will be important to first show that the glucose intolerance phenotype isn't due to a loss of β-cells and/or insulin expression/content prior to investigating the β-cell dysfunction (secretory granules, calcium signaling, etc.).

To fully be able to explain the restricted developmental phenotype, it is necessary to show the expression of *Adrb2* in the developing pancreas, neonate and adult. The authors provide relative values, but this is uninformative. These values should be normalized to a favorite HKG.

Several existing islet/β-cell datasets indicate that *Adrb2* is barely detectable in the adult; the lack of a phenotype in the adult β-cell KO, may be due to this lack of/extremely low expression of *Adrb2*. Also, in the third paragraph of the subsection “Pancreatic Adrb2 is required for glucose homeostasis and insulin secretion in female mice”, it is indicated that *Adrb2* expression is lower in male islets vs. female islets. Since a phenotype is only caused by disruption of *Adrb2* in female embryos/neonates, this comparative expression analysis should be performed at these earlier timepoints.

The authors have nicely shown that disruption of VEGF-A repression and hypervascularization contributes to β-cell dysfunction in the female mice. Are there sex specific differences in VEGF-A repression and/or endothelial density? This will be an important piece of information to understand when/how the sex-specific bias occurs.

The authors claim they observed an increase in intra-islet endothelial density (subsection “Adrb2 suppresses insulin expression and islet vasculature during development”, second paragraph). While Figure 4E suggests that might be the case, intra-islet endothelial density was not measured (but vessel length/ islet area, which may or may not correlate with endothelial density). The authors should directly show intra-islet endothelial density or restate.

The authors show increased insulin expression in neonatal transgenic islets. As Figures 1G and 2A present normalized insulin secretion, it seems the islet insulin content was measured in adult mice. Was islet insulin content measured in neonatal islets (by ELISA)?

As pointed out by the authors, overexpression of VEGF-A and islet hypervascularization increases β-cell proliferation. It has been recently suggested that proliferation renders β-cells immature, and thus less functional (Puri et al., 2018). Thus, increased β-cell proliferation during the neonatal and adult period in the absence of *Adrb2* expression may link hypervascularization with impaired β-cell function. Did the authors analyze β-cell proliferation rate?

Upon observing the significantly impaired insulin secretion from *Adrb2* deficient female mice, the investigators first explore gene expression of genes directly related to β-cell stimulus secretion coupling and find several of these downregulated. However, with the possible exception of *Cacna1c*, these are unlikely to explain the loss of calcium influx in response to KCl-mediated depolarization, which is a quite dramatic effect. Normal insulin expression and impaired stimulus-secretion coupling could explain the increased insulin granule content that is observed, which is an explanation that should be considered or included in the Discussion.

The gene expression analysis suggests there may be a fundamental problem with β-cell maturity/identity, which manifests as a defect in insulin secretion that the investigators observe. A loss of *Neurod1* is reported, but it is a little surprising that this is not followed up in more detail. Additional β-cell maturation markers (*Mafa, Ucn3, Glut2, Pdx1, Nkx6-1*) could all be readily included in the panel of maturity markers assessed by qPCR, and good antibodies are available against all of these that should be used to validate any differences in expression that may be observed. The conclusion that '*Adrb2* is required in neonatal, but not adult, β-cells for insulin secretion and glucose homeostasis' may not accurately describe the phenotype that is observed. It appears instead that *Adrb2* is required for maturation of β-cells during a critical window during postnatal development, which manifests as impaired β-cell function.

[Editors' note: further revisions were requested prior to acceptance, as described below.]

Thank you for resubmitting your work entitled "*Adrb2* controls glucose homeostasis by developmental regulation of pancreatic islet vasculature" for further consideration at *eLife*. Your revised article has been favorably evaluated by Didier Stainier as the Senior Editor, and a Reviewing Editor.

The authors have provided the requested additional experiments and the manuscript has been improved. However, with the addition of new data there is one remaining issue regarding the text that will need to be addressed before acceptance, as outlined below:

With the added data showing there is little to no expression of *Adrb2* in the adult islet, the paragraph describing the adult specific KO doesn't make much sense. Knocking out a gene that isn't expressed would not be expected to cause a phenotype. We are not opposed to the inclusion of the adult data, but the text should be rearranged to make more sense. A possible solution is to move expression data (subsection “*Adrb2* is required in neonatal β-cells for glucose homeostasis and insulin secretion in female mice”, seventh paragraph) up before the adult KO data (sixth paragraph of the aforementioned subsection). The adult phenotype could be introduced as a study to confirm that lack of expression correlated with no phenotype.

---

## [Author Response]

[…] The paper is well written and the data are of high quality and uniformly well presented. With the exception of a number of (relatively minor) issues that are straightforward to address, the conclusions are supported by the data presented. The major concerns are predominantly related to the organization of the manuscript and a couple of important pieces data that are needed to clarify the sex-specific bias and lack of phenotype in the adult specific β-cell knockout.

We appreciate that the Editors and reviewers found the study interesting, convincing, and the data to be of high quality. We have re-organized the Results section and we hope that the changes have improved the narrative. We have also included additional sets of experiments to clarify the lack of metabolic phenotypes with adult β-cell *Adrb2* deletion, as well as differences in *Adrb2* expression and vasculature between male and female islets.

Essential revisions:The organization of the manuscript is difficult to follow. The authors present some of the cKO data first, followed by the i-cKO data, and then back to the cKO data. It would make more sense to combine the cKO data.

The use of the *Adrb2*i-cKO mice allowed us to identify that *Adrb2* is primarily required in neonatal, rather than adult, β-cells for insulin secretion and glucose tolerance.

In the revised manuscript, we combined the metabolic analyses from *Adrb2*i-cKO mice and constitutive *Adrb2*cKO mice and present these results as a revised Figure 1.

For the rest of the studies in the manuscript, we used the non-inducible *Adrb2*cKO mice in Figures 2-4.This was based on the following reasons; (1) We found that *Adrb2* is predominantly expressed in neonatal β-cells (Figure 1H and Figure 1—figure supplement 1K, revised manuscript), (2) the metabolic phenotypes were similar between *Adrb2*i-cKO mice with neonatal Adrb2 deletion and the non-inducible *Adrb2*cKO mice (Figure 1A, B, E, I and J, revised manuscript). An additional reason for mainly using the constitutive *Adrb2*cKO mice was that tamoxifen injections in neonatal *Adrb2*i-cKO mice often resulted in pup mortality, complicating the experiments.

This was the most reasonable organization, that we could think of, to present results from both *Adrb2*cKO and *Adrb2*i-cKO mice, and maintain a coherent narrative. We hope that the Editors/reviewers find this organization to be improved, compared to the initial submission.

And more importantly, it will be important to first show that the glucose intolerance phenotype isn't due to a loss of β-cells and/or insulin expression/content prior to investigating the β-cell dysfunction (secretory granules, calcium signaling, etc.).

We thank the reviewers for this important suggestion. We have now included additional analyses of *adult* islet morphology and insulin content in *Adrb2*cKO mice (Figure 1—figure supplement 1E-J, revised manuscript). We show that adult islet organization, endocrine cell numbers (by immunostaining and counting), and β-cell proliferation (by EdU labeling) were unaffected by *Adrb2* deletion. However, the mutants had a trend toward smaller-sized islets relative to control mice (Figure 1—figure supplement 1H, revised manuscript). Notably, insulin immunoreactivity and ELISA revealed *increased* insulin content in adult *Adrb2*cKO islets (Figure 1—figure supplement 1E, F, revised manuscript), similar to findings in neonatal mutants (Figure 2A and Figure 2—figure supplement 1E, revised manuscript). Together, these results suggest that defects in islet formation and/or maintenance, reduced β-cell numbers, or *diminished* insulin synthesis do not contribute to decreased insulin secretion and glucose intolerance in *Adrb2*cKO mice.

To fully be able to explain the restricted developmental phenotype, it is necessary to show the expression of Adrb2 in the developing pancreas, neonate and adult. The authors provide relative values, but this is uninformative. These values should be normalized to a favorite HKG.Several existing islet/β-cell datasets indicate that Adrb2 is barely detectable in the adult; the lack of a phenotype in the adult β-cell KO, may be due to this lack/extremely low expression of Adrb2. Also, in the third paragraph of the subsection “Pancreatic Adrb2 is required for glucose homeostasis and insulin secretion in female mice”, it is indicated that Adrb2 expression is lower in male islets vs. female islets. Since a phenotype is only caused by disruption of Adrb2 in female embryos/neonates, this comparative expression analysis should be performed at these earlier timepoints.

We analyzed *Adrb2* transcripts in male and female islets at neonatal stages (P2 and P6) and adults (P60) using qPCR analyses. To clarify, we always normalize each transcript to 18S rRNA and then express the fold-change in levels relative to a set-point (particular time-point, control versus mutant etc.). We have clearly indicated this in the revised figure legends.

We found that *Adrb2* expression was significantly decreased in adult islets compared to neonatal stages (Figure 1H, revised manuscript). The decline in *Adrb2* expression was observed in both female and male islets. Moreover, *Adrb2* levels were significantly higher in female islets compared to males at all the time-points assessed (Figure 1H, revised manuscript).

These results support the point made by the reviewers that the lack of metabolic phenotypes with adult β-cell-specific *Adrb2* deletion could be due to lower *Adrb2* expression at this stage.

We have stated as such in subsection “*Adrb2* is required in neonatal β-cells for glucose homeostasis and insulin secretion in female mice”, seventh paragraph.

The authors have nicely shown that disruption of VEGF-A repression and hypervascularization contributes to β-cell dysfunction in the female mice. Are there sex specific differences in VEGF-A repression and/or endothelial density? This will be an important piece of information to understand when/how the sex-specific bias occurs.

We assessed pancreatic *Vegfa* expression and islet vasculature in neonatal male and female control mice at postnatal day 6 (P6), using qPCR analysis and PECAM1 immunohistochemistry, respectively. We observed higher *Vegfa* expression and enhanced intra-islet vasculature in neonatal males, compared to females. These new results are shown in (Figure 2—figure supplement 1K-M, revised manuscript).

These data, together with the findings of sexually dimorphic *Adrb2* expression in neonatal islets (Figure 1H, revised manuscript), suggest that there are sex-specific differences in *Adrb2*-mediated repression of *Vegfa* expression and intra-islet vasculature that are initiated during development, at least as early as P2.

The authors claim they observed an increase in intra-islet endothelial density (subsection “Adrb2 suppresses insulin expression and islet vasculature during development”, second paragraph). While Figure 4E suggests that might be the case, intra-islet endothelial density was not measured (but vessel length/ islet area, which may or may not correlate with endothelial density). The authors should directly show intra-islet endothelial density or restate.

We have modified the text to state “these analyses revealed a prominent increase in intra-islet vasculature in *Adrb2*cKO islets”.

The authors show increased insulin expression in neonatal transgenic islets. As Figures 1G and 2A present normalized insulin secretion, it seems the islet insulin content was measured in adult mice. Was islet insulin content measured in neonatal islets (by ELISA)?

We performed additional experiments to measure insulin content from neonatal islets at P6 using ELISA. We observed a significant 2-fold increase in insulin content in neonatal *Adrb2* cKO islets (Figure 2—figure supplement 1E, revised manuscript), similar to the results from adult mutant islets (Figure 1—figure supplement 1F, revised manuscript).

As pointed out by the authors, overexpression of VEGF-A and islet hypervascularization increases β-cell proliferation. It has been recently suggested that proliferation renders β-cells immature, and thus less functional (Puri et al., 2018). Thus, increased β-cell proliferation during the neonatal and adult period in the absence of Adrb2 expression may link hypervascularization with impaired β-cell function. Did the authors analyze β-cell proliferation rate?

To assess β-cell proliferation, we performed EdU labeling in conjunction with insulin immunostaining in neonatal and adult mice at P6 and 2 months of age, respectively. There were no significant differences in β-cell proliferation between control and *Adrb2*cKO islets at either age (Figure 1—figure supplement 1I, J for adults and Figure 2—figure supplement 1C,D for neonates, revised manuscript). These results are consistent with the findings that β-cell numbers are unaffected by Adrb2 loss in neonatal and adult animals (Figure 1—figure supplement 1G for adults and Figure 2—figure supplement 1A for neonates, revised manuscript).Of note, although Adrb2 loss did not elicit enhanced adult β-cell proliferation, postulated to be a characteristic of juvenile β-cells (Puri et al., 2018), we did observe dysregulated expression of specific β-cell genes associated with β-cell maturation, including *Neurod1, Glut2* and *Npy* in adult *Adrb2*cKO islets (see our last response below).

Upon observing the significantly impaired insulin secretion from Adrb2 deficient female mice, the investigators first explore gene expression of genes directly related to β-cell stimulus secretion coupling and find several of these downregulated. However, with the possible exception of Cacna1c, these are unlikely to explain the loss of calcium influx in response to KCl-mediated depolarization, which is a quite dramatic effect. Normal insulin expression and impaired stimulus-secretion coupling could explain the increased insulin granule content that is observed, which is an explanation that should be considered or included in the Discussion.

The increased insulin granule content observed in *Adrb2* mutant β-cells could result from impaired secretion. However, insulin transcript levels were also elevated in *Adrb2*cKO islets, suggesting that enhanced synthesis could also be an important contributing factor to the elevated insulin content in *Adrb2*cKO islets. We have modified the text in the Discussion accordingly (third paragraph).

The gene expression analysis suggests there may be a fundamental problem with β-cell maturity/identity, which manifests as a defect in insulin secretion that the investigators observe. A loss of Neurod1 is reported, but it is a little surprising that this is not followed up in more detail. Additional β-cell maturation markers (Mafa, Ucn3, Glut2, Pdx1, Nkx6-1) could all be readily included in the panel of maturity markers assessed by qPCR, and good antibodies are available against all of these that should be used to validate any differences in expression that may be observed. The conclusion that 'Adrb2 is required in neonatal, but not adult, β-cells for insulin secretion and glucose homeostasis' may not accurately describe the phenotype that is observed. It appears instead that Adrb2 is required for maturation of β-cells during a critical window during postnatal development, which manifests as impaired β-cell function.

We thank the Editors/reviewers for the suggestion. To address if *Adrb2* loss affects β-cell maturation, we analyzed additional genes associated with functional maturity including *Ucn3, Glut2 (Slc2a2), Pdx1, Nkx6.1*, and*Npy* using qPCR. We observed a marked up-regulation of *Npy* and decreased *Glut2* in adult Adrb2 cKO islets, while *Ucn3 Pdx1,* and *Nkx6.1* were unaffected (Figure 3F, revised manuscript). The changes in NPY and Glut2 expression in Adrb2 cKO islets were supported by immunostaining analyses (Figure 3—figure supplement 1G, revised manuscript). The decreased expression of *Neurod1* and *Glut2* and up-regulation of *Npy* in adult *Adrb2*cKO islets, which are all well-established features of juvenile β-cells, could alter β-cell maturation and/or identity and eventually contribute to the observed secretory dysfunction in mutant islets. We have modified the text in the Discussion accordingly (fourth paragraph).

[Editors' note: further revisions were requested prior to acceptance, as described below.]

The authors have provided the requested additional experiments and the manuscript has been improved. However, with the addition of new data there is one remaining issue regarding the text that will need to be addressed before acceptance, as outlined below:With the added data showing there is little to no expression of Adrb2 in the adult islet, the paragraph describing the adult specific KO doesn't make much sense. Knocking out a gene that isn't expressed would not be expected to cause a phenotype. We are not opposed to the inclusion of the adult data, but the text should be rearranged to make more sense. A possible solution is to move expression data (subsection “Adrb2 is required in neonatal β-cells for glucose homeostasis and insulin secretion in female mice”, seventh paragraph) up before the adult KO data (sixth paragraph of the aforementioned subsection). The adult phenotype could be introduced as a study to confirm that lack of expression correlated with no phenotype.

We have re-arranged the text in the subsection “*Adrb2* is required in neonatal β-cells for glucose homeostasis and insulin secretion in female mice” to discuss the *Adrb2* expression data and effects of neonatal β-cell deletion *before* the adult deletion phenotypes.

We have also re-arranged Figure 1—figure supplement 1 and figure legend to be consistent with the changes in the manuscript file.